# Unveiling the Secret Recipe:
# A Guide For Supervised Fine-Tuning Small LLMs

**Aldo Pareja**[1,2*], **Nikhil Shivakumar Nayak**[1,2], **Hao Wang**[1,2], **Krishnateja Killamsetty**[3],
**Shivchander Sudalairaj**[1,2], **Wenlong Zhao**[1,5†], **Seungwook Han**[1,4†],
**Abhishek Bhandwaldar**[1,2], **Guangxuan Xu**[1,2], **Kai Xu**[1,2], **Ligong Han**[1,2], **Luke Inglis**[2,3],
**Akash Srivastava**[1,2]

[1]Red Hat AI Innovation    [2]MIT-IBM Watson AI Lab    [3]IBM Research
[4]Massachusetts Institute of Technology    [5]University of Massachusetts Amherst

## ABSTRACT

The rise of large language models (LLMs) has created a significant disparity: industrial research labs with their computational resources, expert teams, and advanced infrastructures, can effectively fine-tune LLMs, while individual developers and small organizations face barriers due to limited resources to effectively explore the experiment space. In this paper, we aim to bridge this gap by presenting a comprehensive study on supervised fine-tuning of LLMs using instruction-tuning datasets spanning diverse knowledge domains and skills. We focus on small-sized LLMs (3B to 7B parameters) for their cost-efficiency and accessibility. We explore various training configurations and strategies across four open-source pre-trained models. We provide detailed documentation of these configurations, revealing findings that challenge several common training practices, including hyperparameter recommendations from TULU (Wang et al., 2023b) and phased training recommended by Orca (Mitra et al., 2023). The code used for the experiments can be found here: https://github.com/instructlab/training.

Key insights from our work include: (i) larger batch sizes paired with lower learning rates lead to improved model performance on benchmarks such as MMLU, MTBench, and Open LLM Leaderboard; (ii) early-stage training dynamics, such as lower gradient norms and higher loss values, are strong indicators of better final model performance, allowing for early termination of sub-optimal runs and significant computational savings; (iii) through a thorough exploration of hyperparameters like warmup steps and learning rate schedules, we provide guidance for practitioners and find that certain simplifications do not compromise performance; and (iv) we observe no significant difference in performance between phased (sequentially training on data divided into phases) and stacked (training on the entire dataset at once) strategies, but stacked training is simpler and more sample efficient. With these findings holding robustly across datasets as well as model families and sizes, we hope this study serves as a guide for practitioners fine-tuning small LLMs and promotes a more inclusive research environment for LLM development.

## 1 INTRODUCTION

Large language models (LLMs) are growing in size, but bigger is not always better. Small-sized LLMs (3B to 7B parameters) become increasingly popular among developers with limited resources and are emerging as the backbone of enterprise AI systems due to their adaptability and efficiency

---

*  Correspondence to: Aldo Pareja <agparejac@gmail.com>.
†  Work done while interning at Red Hat.

(RedHat, 2024; Zhang et al., 2023; Lee, 2024). Compared to larger LLMs, fine-tuning and deploying these models is faster, more cost-effective, and does not require specialized infrastructure or extensive hardware like GPUs and TPUs. Moreover, small-sized LLMs can be customized with domain-specific data, enabling them to address specific tasks, domains, or organizational needs while achieving performance comparable to, or even exceeding, proprietary LLMs in specialized areas. Finally, they can be hosted on consumer-grade machines, which offer individual developers and organizations full control over their data and fine-tuned models, reducing the risk of data breaches or non-compliance with regulations such as GDPR (2016).

Instruction tuning plays a pivotal role in unlocking the potential of small-sized LLMs. It enables these models to follow user instructions, improves zero-shot capabilities, and customizes them as domain-specific experts (Ouyang et al., 2022; Chung et al., 2022; Scao et al., 2023). Among the various types of instruction-tuning data, knowledge and skills datasets are particularly important. As defined in our previous work (Sudalairaj et al., 2024), knowledge datasets focus on factual accuracy across diverse domains, while skills datasets emphasize foundational and compositional abilities such as reasoning, coding, and problem-solving. These datasets are more accessible, often of higher quality, and exhibit less biases compared to other sources (Gunasekar et al., 2023; Longpre et al., 2023; Wang et al., 2023a; Ding et al., 2023b). Furthermore, they contribute to improved model memorization, reduced hallucinations, and enhanced reasoning abilities (Zhou et al., 2023). The diversity within such datasets also fosters better model generalization across tasks (Wei et al., 2021; Sanh et al., 2022; Wei et al., 2022). Although significant efforts have been made to generate large-scale knowledge and skills instruction datasets (Sudalairaj et al., 2024; Wang et al., 2022; Taori et al., 2023; Xu et al., 2023; Li et al., 2024) and many open-source instruction-tuned models are now available, there is limited research on how to effectively fine-tune base models from scratch.

Practitioners have limited resources to reference when searching for optimal training strategies and hyper-parameters for instruction-tuning small LLMs on knowledge and skills data. Many LLMs are closed-source, and even those that are open-source often lack detailed technical reports describing how to set up hyper-parameters or which configurations were attempted but unsuccessful. As a result, critical factors like batch size and learning rate, as well as their impact on final model performance, remain unclear. Additionally, phase training is increasingly used for instruction tuning, where LLMs are fine-tuned progressively, starting with simple instruction-following data (e.g., general knowledge from elementary or middle school), then moving to foundational knowledge (e.g., graduate-level content), and finally to skills-based data. However, it is unclear how well phase training outperforms traditional stacked training where all data is combined into a single phase. Identifying an effective set of hyper-parameters is especially difficult for users with limited computational resources. This motivates the main question we aim to study:

*How can we effectively fine-tune a small-size LLM (3B–7B parameters) on instruction tuning datasets that cover diverse knowledge and skills?*

In this paper, we present a comprehensive empirical study on supervised fine-tuning small-size LLMs and compare our findings with existing research on this topic. We experiment with 4 open-source models—`Granite 3B`, `Granite 7B`, `Llama 3.2 3B` and `Mistral 7B`—fine-tuning them on five datasets: an instruction-following dataset with 308,343 samples, a foundational knowledge dataset with 231,178 samples, a complex skills dataset with 285,966 samples, the TULU mixture v2 dataset, and a domain-specific math, reasoning and coding dataset. Through a series of experiments, we systematically vary hyper-parameters and training strategies and collect experimental results. Our findings challenge several widely accepted practices, including those recommended by TULU (Wang et al., 2023b; Ivison et al., 2023), which is often considered a gold standard for LLM fine-tuning. For example, they use a (small) batch size of 128 samples, which we find to underperform in our experiments. We conjecture that this choice was driven by their computational constraints, as larger batch sizes can produce models with higher downstream performance but require much longer training time under limited computing resources. Additionally, while learning rate schedulers with warm-up and decay are widely used in neural network training, including in TULU, our results show that these techniques have minimal impact on model downstream performance.

Our key observations are: (i) larger batch sizes combined with lower learning rates improve generalization and performance on benchmarks like MMLU (Hendrycks et al., 2020), MTBench (Zheng et al., 2023), and Open LLM Leaderboard v2; (ii) early-stage training dynamics, such as lower gra-

dient norms and higher loss values, are strong indicators of final model performance, enabling early termination of sub-optimal runs and significant computational savings; (iii) omitting warmup steps and using constant learning rates does not compromise performance; and (iv) stacked training offers similar performance to phased training but is more sample efficient. We also address adaptations for new architectures and emphasize the importance of efficient data handling techniques, such as bucketing and balanced compute distribution across GPUs. Our findings aim to provide practitioners with actionable insights to fine-tune LLMs more effectively, optimizing performance while simplifying the training process. This can benefit the open-source community focused on instruction tuning and serve as a reference for practitioners with limited computational resources.

RELATED WORK

**Instruction Tuning Data.** Instruction tuning with diverse, large-scale datasets can effectively improve LLM performance across downstream tasks (Wang et al., 2023c; Honovich et al., 2023; Chung et al., 2024; Isik et al., 2024; Cheng et al., 2024). Recent studies have found that large-scale instruction tuning data focusing on knowledge and skills is particularly beneficial for adapting LLMs to customized domains or applications, improving factual recall and reducing hallucinations (Cheng et al., 2023; Allen-Zhu & Li, 2023; Yang et al., 2024). This observation has led to a growing body of research introducing novel instruction tuning datasets. For instance, several works leveraged larger, more powerful LLMs (e.g., ChatGPT (OpenAI, 2023; 2022) and Mistral models (Jiang et al., 2023; 2024)) to distill instruction tuning data from them using seed examples provided by users (Mitra et al., 2024; Xu et al., 2023; Ding et al., 2023a; Peng et al., 2023; Mukherjee et al., 2023). GLAN (Li et al., 2024) and LAB (Sudalairaj et al., 2024) further advanced this area by proposing taxonomy-driven frameworks to enhance the diversity of synthetic instruction tuning data. Building on these datasets, many studies explored strategies to optimize dataset composition, select representative data subsets, and evaluate data quality before incorporating them into model training (Ivison et al., 2023; Liu et al., 2023; Li et al., 2023; Xie et al., 2023). While these advancements have driven rapid progress in instruction-tuned LLMs, limited work has focused on how to effectively use such data during training to achieve optimal performance, or how training outcomes vary with different compute budgets (e.g., GPUs and TPUs). In this paper, we fill this gap by conducting an extensive set of experiments to investigate various training strategies and hyperparameters for customizing small LLMs on these datasets, analyzing how different configurations interact with available compute resources to impact the downstream performance of fine-tuned models.

**Training Dynamics.** Training configurations and hyper-parameter setups play a pivotal role in training LLMs, as they directly influence model performance, convergence stability, and resource efficiency. Most research has focused on the pre-training phase, as it is the most resource-intensive part of LLM development (Yang et al., 2022; Hägele et al., 2024; Bi et al., 2024; Kaplan et al., 2020; Rosenfeld et al., 2019; Gunter et al., 2024; Dubey et al., 2024). For example, Sardana et al. (2024); Hoffmann et al. (2022) introduced scaling laws to determine optimal model sizes for given datasets and Hägele et al. (2024) proposed novel learning rate schedulers as alternatives to conventional cosine decay. Additionally, recent research proposed to incorporate instruction tuning data alongside pre-training data as part of a decay phase in pre-training, linking to the body of research on continual pre-training (Hu et al., 2024; Ibrahim et al., 2024; Lesort et al., 2021; Scialom et al., 2022). In contrast, our work shifts the focus to customizing pre-trained LLMs through instruction tuning, highlighting under-explored challenges in training strategies and hyper-parameter configurations for this stage. Many instruction tuning studies either omit the reporting of hyperparameters altogether (Mukherjee et al., 2023) or only provide a selective set of hyperparameters used in successful runs (Wang et al., 2023b; Ivison et al., 2023; Xu et al., 2023), often without disclosing failed experiments or alternative configurations explored during their research. In contrast, we conduct extensive experiments, exploring a range of hyper-parameters and training strategies. Our findings challenge several widely used practices, including TULU, and we hope that our work can serve as a valuable reference for practitioners and spark discussions on a deeper understanding of training dynamics for fine-tuning LLMs.

**Traditional Wisdom in Neural Networks Training.** Identifying effective training configurations to improve model generalization has been an active area of research long before the rise of LLMs (Zhang et al., 2017; Srivastava et al., 2014; Ioffe & Szegedy, 2015). For example, Jiang et al.

(2019) conducted extensive experiments to analyze how different generalization measures predict final model performance, offering insights for hyper-parameter tuning. However, many established findings do not always extend to LLMs. For example, Keskar et al. (2016) found that large batch sizes led to poor generalization due to sharp minima. In contrast, our work shows that using large batches results in higher scores on MT-Bench, indicating improved generalization on downstream performance. This discrepancy arises due to the difference in experimental settings, where both architecture (CNNs and MLPs vs. Transformers), and task domains (CV vs. NLP) vary significantly; and importantly LLMs are pre-trained on massive datasets which drastically changes the starting point for supervised fine tuning (Peng et al., 2023). Additionally, fine-tuning LLMs poses unique challenges, as it often requires state-of-the-art clusters spanning multiple machines, each equipped with multiple GPUs, and advanced networking to optimize speed, memory efficiency, and scalability using frameworks such as Deepspeed (Rasley et al., 2020), PyTorch's FSDP (Zhao et al., 2023) or Megatron-LM (Narayanan et al., 2021). These requirements are not typically encountered in conventional deep learning workflows.

## 2  EXPERIMENTAL SETUP

This section outlines the pre-trained LLMs, the datasets curated for fine-tuning these models, the training strategies used, and the hyper-parameters tested in our experiments. Details on the training infrastructure and optimization techniques used in our experiments can be found in Appendix A.4. We directly present the experiments and results in the following sections. For readers interested in the detailed experimental design and hypotheses, please refer to Appendix A.5.

### 2.1  BASE MODELS AND DATASETS

We conduct experiments using four open-source, small-sized LLMs: `Granite 3B`[1], `Granite 7B`, `Mistral 7B`, and `LLaMA 3B`. The Granite models (Mishra et al., 2024), developed by IBM Research, are decoder-only architectures designed for enterprise applications, with the "3B" and "7B" designations indicating their parameter counts of 3 billion and 7 billion, respectively. The Mistral 7B model (Jiang et al., 2023), created by Mistral AI, is a dense, decoder-only transformer model with 7 billion parameters, optimized for high performance relative to its size. While our primary focus is on the Granite and Mistral models, given their permissive Apache-2.0 licensing, we include the LLaMA model (Touvron et al., 2023) in specific experiments to test the generalizability of our findings. These experiments further validating the robustness of our conclusions across architectures and model sizes within the small-sized LLM category.

We curated a comprehensive dataset designed to progressively enhance the base models' capabilities in instruction following (phase 00), foundational knowledge (phase 05), and complex skills (phase 10) (see Sudalairaj et al., 2024, for details). This dataset is organized into three phases, each targeting specific aspects of language understanding and generation (see Appendix A.3). We also explored an alternative dataset partitioning based on task difficulty, where phases are defined by sentence length as a proxy for difficulty (further detailed in Appendix A.6.1). We also conducted experiments with the TULU dataset (Wang et al., 2023b; Ivison et al., 2023), a diverse mix of complex, instruction-tuning data from human and GPT-4 sources; details are provided in the main results section. Finally, we test our findings on a synthetically generated Math, Reasoning, and Code dataset, similar to our other datasets, with a focus on tasks in these domains to ensure they hold for domain-specific datasets.

### 2.2  TRAINING STRATEGIES

We explore two training strategies—*sequential phased training* and *stacked training*. Phased training follows the approach adopted by recent instruction tuning research (Sudalairaj et al., 2024; Mitra et al., 2023; Pang et al., 2024), where the base model is fine-tuned on different data types in a pre-determined sequence. This strategy aims to mitigate catastrophic forgetting and allows the model to build progressively on knowledge and skills acquired in earlier stages. In our experiments, models are fine-tuned in multiple phases, each focusing on a specific type of data (see Appendix A.3 for details on the datasets used in each phase). At the end of each phase, the best-performing checkpoint is

---

[1] We got early access to a preview version of the Granite 3B model.

Table 1: Summary of hyperparameter configurations.

| Hyperparameter | TULU | TULU++ | LAB |
|---|---|---|---|
| **Effective Batch Size** | 128 samples | Same as TULU | 3,840 or 7,680 samples |
| **Learning Rate Scheduler** | Warmup ratio: 0.03 
 Linear decay until the end of training | Same as TULU | Warmup ratio: 0.01 (25 steps linear warmup) 
 No decay (constant rate after warmup) |
| **Number of Epochs** | 3 | 4 | 10 |
| **Goal Learning Rate** | $2 \times 10^{-5}$ | $3 \times 10^{-5}$ | $2 \times 10^{-5}$ (also tested with higher rates) |

selected based on evaluation metrics before proceeding to the next phase. Stacked training combines all data from different phases into a single training phase, exposing the model to diverse data simultaneously. This approach simplifies the training pipeline by eliminating the need for phase-wise data curation.

## 2.3 HYPERPARAMETERS

Our experiments explore various hyperparameter configurations to analyze their impact on training dynamics and model performance.

- **Batch Size.** We investigate effective batch sizes of 128 (small), 3,840 (medium), and 7,680 (large) samples. The effective batch size is achieved through a combination of micro-batch sizes and gradient accumulation steps. For instance, on 64 GPUs, we can process a batch of 3,840 samples in a single micro-batch, whereas on 1 GPU or 8 GPUs, we use gradient accumulation to approximate the same batch size. We confirm that gradient accumulation on a single node produces equivalent results to multi-node distributed training, with details in Appendix A.6.10.

- **Learning Rate and Warmup Steps.** We experiment with various goal learning rates: $1 \times 10^{-6}$, $5 \times 10^{-6}$, $2 \times 10^{-5}$, $3 \times 10^{-5}$, $4 \times 10^{-5}$, $6 \times 10^{-5}$, $8 \times 10^{-5}$, and $1 \times 10^{-4}$. Warmup steps are varied among 0, 25, and 100, corresponding to different numbers of samples processed before reaching the goal learning rate. The learning rate schedule typically involve a linear warmup to the goal learning rate, followed by either a constant learning rate or the cosine decay schedule.

- **Training Configurations.** We consider three main hyperparameters configurations: LAB (Sudalairaj et al., 2024), TULU (Wang et al., 2023b; Ivison et al., 2023), and a new configuration introduced in this paper, TULU++. Details of these configurations are provided in Table 1.

We used the LAB hyperparameter configuration for all experiments where we varied a single factor (e.g., batch size, learning rate, learning rate schedule, training strategy) while keeping all other settings constant to isolate its effect. For comparisons between TULU and LAB, we directly used the respective configurations. LAB and TULU were chosen as primary configurations due to their prominence: TULU is widely regarded as a gold standard for fine-tuning LLMs with high-quality instruction datasets, while LAB introduces a multi-phase tuning framework leveraging knowledge and skills data to reduce reliance on human annotations. This dual comparison allowed us to systematically evaluate established practices and propose actionable guidelines.

## 2.4 EVALUATION METRICS

**Benchmarks.** To assess the models' performance and ability to generalize, we use two primary benchmarks: MMLU (Hendrycks et al., 2020) and MTBench (Zheng et al., 2023). MMLU assesses the models' knowledge and reasoning across a wide range of subjects. It includes questions from 57 subjects spanning STEM, humanities, social sciences, and more, testing the model's ability to recall factual knowledge and apply reasoning skills to answer multiple-choice questions. MTBench evaluates multi-turn conversational abilities and generalization to unseen tasks. It measures the quality of responses in dialogue settings, focusing on coherence, relevance, informativeness, and adherence to instructions. The benchmark covers diverse tasks such as reasoning, coding, mathematics, and other skill-based domains. Additionally, we evaluated our models on MMLU-Pro, GPQA, MuSR, MATH, IFEval, and BBH from the Open LLM Leaderboard v2[2]. For the comparison with the TULU dataset, we used the same benchmarks as in the TULU paper (Wang et al., 2023b; Ivison

---

[2]https://huggingface.co/docs/leaderboards/open_llm_leaderboard/about

et al., 2023): MMLU, GSM8K, BBH, ToxiGen, and TruthfulQA; details of this evaluation are provided in the TULU vs. LAB section in the main results. Finally, we also include evaluations on ARC (Clark et al., 2018) and GSM8K (Cobbe et al., 2021).

**Efficiency Metrics.** "Number of Samples" is defined as the product of the number of gradient updates and the batch size (i.e., #samples = #gradient updates × BS), reflecting sample efficiency by indicating the amount of data processed to achieve a certain performance level. In our experiments, *sample efficiency* and *compute efficiency* effectively represent the same metric. This is because, whether we use multiple GPUs for faster fine-tuning or a single GPU, the total computational workload (measured in GPU-hours) remains the same. Using multiple GPUs reduces the wall-clock training time by parallelizing computations across devices, but the total number of computational operations performed remains unchanged. The use of multiple GPUs or a single GPU with gradient accumulation are equivalent techniques and do not impact overall performance. Thus, methods that are more sample-efficient (i.e., they use fewer data samples) also exhibit better compute efficiency, since they require fewer total computations and, consequently, less aggregate training time to achieve the same performance, independent of the hardware configuration.

## 3 MAIN RESULTS

In this section, we present the empirical findings of our experiments, focusing on the impact of different training strategies, batch sizes, and hyperparameter configurations on the fine-tuning performance of LLMs. We present results using the Granite 7B model and provide additional experiments to other model sizes and architectures (Granite 3B, LLaMA 3B, and Mistral 7B models) in Appendix A.6.8 to validate the robustness and generalizability of our findings. Additionally, we conducted experiments on a domain-specific Math, Reasoning, and Code (MRC) dataset to evaluate the applicability of our findings to specialized fine-tuning scenarios, with further details and results provided in Appendix A.6.7. We include baseline scores for the Granite and LLaMA base pretrained models in applicable tables to facilitate easier interpretation of fine-tuned performance. MTBench scores are not provided for baseline models, as these benchmarks evaluate instruction-following and conversational capabilities not present in base models.

### 3.1 STACKED TRAINING VS. SEQUENTIAL PHASED TRAINING

We conducted a comprehensive comparison between *stacked training* and *sequential phased training* to evaluate their effectiveness in fine-tuning small sized LLMs. The analysis was performed using the Granite 7B model and evaluated on the MMLU, MTBench, ARC, GSM8K, and Leaderboard (BBH, MATH, MuSR) benchmarks. We observed that stacked training slightly outperformed sequential phased training and is more sample efficient across all batch sizes – 128 and 3,840. The detailed comparison of performance across batch sizes is presented in Appendix A.6.1, along with corresponding figures.

Table 2: Comparison of Stacked vs. Phased Training Strategies. Samples indicate the number of data points required to reach peak performance for each benchmark. Cells highlighted in green indicate better scores, and blue indicates higher sample efficiency (fewer samples used).

| Benchmark | Score | | | Samples | |
|---|---|---|---|---|---|
| | **Granite Base** | **Stacked** | **Phased** | **Stacked** | **Phased** |
| **MMLU** | 0.48 | 0.53 | 0.52 | 3,694,080 | 7,859,902 |
| **MTBench** | - | 6.77 | 6.76 | 4,392,960 | 8,057,918 |
| **Leaderboard (BBH)** | 0.09 | 0.10 | 0.10 | 3,694,080 | 8,057,918 |
| **Leaderboard (MATH Lvl 5)** | 0.01 | 0.01 | 0.00 | 3,694,080 | 8,057,918 |
| **Leaderboard (MuSR)** | 0.01 | 0.08 | 0.07 | 3,694,080 | 8,057,918 |
| **ARC** | 0.78 | 0.76 | 0.74 | 3,694,080 | 8,057,918 |
| **GSM8K** | 0.11 | 0.39 | 0.37 | 3,694,080 | 8,057,918 |

As shown in Table 2, we compare the performance of stacked and phased training strategies using the LAB hyperparameter configuration, which provided the best overall results for both approaches.

Stacked training achieves slightly better performance on most benchmarks and comparable performance on the rest, while also being more sample-efficient, requiring significantly fewer samples to reach peak performance. Detailed plots and scores over all checkpoints during training are provided in Appendix A.6.1.

These findings suggest that the stacked training approach improves performance by enabling the model to learn from diverse data simultaneously. Additionally, phased training demands extra time and samples to identify the optimal checkpoint for transitioning between phases. This requires running the model longer to determine peak performance. The increased overhead further diminishes the sample efficiency of phased training compared to the stacked approach.

## 3.2 IMPACT OF BATCH SIZE

We investigated the effect of batch size on model performance by experimenting with effective batch sizes of 128, 3,840, and 7,680 samples. The experiments were conducted using the Granite 7B model and evaluated on the MMLU and MTBench benchmarks. To ensure a fair comparison, we ran each experiment for approximately the same number of gradient steps.

**Observations.** Larger batch sizes lead to better final performance but may require more computational resources and training samples. For stacked training, larger batch sizes uniformly resulted in improved performance on both MMLU and MTBench. The performance gains observed with larger batch sizes can be attributed to reduced statistical error in gradient estimation during optimization. Since each gradient step is computed by averaging over more training examples, the variance of the estimate decreases proportionally to $\frac{1}{\sqrt{n}}$ (where $n$ is the batch size). This reduction in estimation noise enables more consistent parameter updates that better align with the true gradient of the loss function, potentially leading to more efficient optimization trajectories. This reduced noise in the gradients likely enables the optimization process to progress toward a minimum with fewer fluctuations, allowing the model to settle near the pre-trained parameters (Hoffer et al., 2017). We hypothesize that this effect minimizes deviation from the pre-trained state, as also observed by Keskar et al. (2016), helping the model adapt to new data without significant departure from its initial configuration, thereby reducing forgetting. For phased training, the batch size of 3,840 samples outperformed the smaller batch size of 128 samples. While larger batch sizes still yield better overall performance, the impact is less pronounced compared to stacked training. This could be because, in phased training, each phase focuses on a specific type of data, resulting in batches where the data is more homogeneous; therefore, the benefits of reduced gradient noise from larger batch sizes are less significant, and the impact of larger batch sizes is less pronounced compared to stacked training. Table 3 illustrates the performance of different batch sizes in both stacked and phased training on the MMLU and MTBench benchmarks.

Table 3: Comparison of Batch Sizes Across Stacked and Phased Training Strategies on MMLU and MTBench Benchmarks. Green cells indicate better scores, while blue cells highlight higher sample efficiency (fewer samples required).

| Benchmark | Strategy | Score | | | | Samples | | |
|---|---|---|---|---|---|---|---|---|
| | | Granite Base | 128 | 4K | 8K | 128 | 4K | 8K |
| MMLU | Stacked | 0.48 | 0.516 | 0.526 | 0.529 | 2,099,328 | 3,694,080 | 8,885,760 |
| | Phased | 0.48 | 0.513 | 0.524 | - | 2,915,233 | 7,859,902 | - |
| MTBench | Stacked | - | 6.406 | 6.768 | 6.831 | 1,799,424 | 4,392,960 | 8,586,240 |
| | Phased | - | 6.325 | 6.756 | - | 2,815,265 | 8,057,918 | - |

**Trade-off.** We observed that models trained with smaller batch sizes achieved higher performance faster in terms of the number of processed samples but plateaued earlier compared to those trained with larger batch sizes. Conversely, models with larger batch sizes required longer training time to reach similar performance levels due to fewer gradient updates per epoch. This trend is illustrated in Appendix A.6.2, where the performance curves for larger batch sizes span a greater number of training samples. However, with extended training, larger batch sizes led to higher final performance.

## 3.3 EFFECT OF LEARNING RATE SCHEDULES ON LARGE BATCH SIZES

We explored whether using a cosine decay learning rate schedule improves model performance when training with large batch sizes. Cosine decay is often thought to facilitate convergence by allowing higher initial learning rates and gradually reducing them. It can be particularly beneficial when training with large batches that may require larger steps to make meaningful progress. We conducted experiments using the Granite 7B model with an effective batch size of 3,840 samples. We compared two learning rate schedules: a constant learning rate and a cosine decay schedule. The learning rate tested was $2 \times 10^{-5}$.

Table 4: Effect of Cosine Decay on MMLU and MTBench Scores at Learning Rate $2 \times 10^{-5}$. Cells highlighted in green indicate better scores, and blue indicates higher sample efficiency (fewer samples used).

| Benchmark | Score | | | Samples | |
|---|---|---|---|---|---|
| | Granite Base | No Decay | Cosine Decay | No Decay | Cosine Decay |
| **MMLU** | 0.48 | 0.5242 | 0.5251 | 2,475,200 | 1,188,096 |
| **MTBench** | - | 6.7562 | 6.6813 | 2,673,216 | 1,188,096 |

**Observations.** As shown in Table 4, the models trained with a constant learning rate (no decay) performed on par with those trained with a cosine decay schedule on MMLU and even outperformed them on MTBench. Detailed plots are provided in Appendix A.6.3.

**Analysis.** Our findings suggest that, contrary to common practice, cosine decay may not improve model performance when fine-tuning small-size LLMs with large batch sizes. Instead, a constant learning rate ensures consistent progress throughout training, under the assumption that the initial rate is suitable for stable training. For practitioners, this implies that using a constant learning rate could simplify the training process without compromising performance, and may even offer slight improvements.

## 3.4 TULU VS. LAB

We compared the TULU and LAB hyperparameter configurations to assess their effectiveness in enhancing the model's memorization and generalization capabilities. Memorization was evaluated using a subset of the MMLU benchmark focused on factual knowledge domains, while generalization was assessed using the MTBench benchmark, which tests the model's ability to perform diverse and complex tasks requiring various skills. Detailed plots and scores over all checkpoints during training are provided in Appendix A.6.4, along with performance results for the Leaderboard (BBH, MATH Lvl 5, MuSR), ARC, and GSM8K benchmarks. Tables 8 and 9 show that LAB outperforms TULU across all benchmarks.

Table 5: Evaluation results for TULU Dataset across batch sizes. Cells highlighted in green indicate better scores. "Ours" refers to the configuration with a batch size of 4k, while "Theirs" uses the original batch size of 128.

| Benchmark | Theirs (128 Batch Size) | Ours (4k Batch Size) |
|---|---|---|
| **MMLU** | 0.48 | 0.50 |
| **BBH** | 0.40 | 0.44 |
| **GSM8K** | 0.25 | 0.28 |
| **ToxiGen** | 0.54 | 0.55 |
| **TruthfulQA** | 0.45 | 0.44 |

**Cross-Dataset Evaluation with the TULU Dataset.** To further investigate the impact of batch size on fine-tuning performance, we conducted an experiment using the TULU dataset (Wang et al., 2023b; Ivison et al., 2023). This dataset is a refined mixture of instruction-tuning data, integrating both human and GPT-4-generated instructions that are complex and cover various domains. We

fine-tuned the Granite 7B model on the TULU dataset using two configurations: batch size of 128 as recommended by TULU, and our configuration with a larger batch size of 3,840 (denoted as 4k). The rationale for testing across datasets was to determine whether the advantages of larger batch sizes observed on our datasets would generalize to different fine-tuning datasets. We evaluated the models using the same benchmarks as in the TULU paper: MMLU (Hendrycks et al., 2020), GSM8K (Cobbe et al., 2021), BBH (Suzgun et al., 2022), ToxiGen (Hartvigsen et al., 2022), and TruthfulQA (Lin et al., 2021). The results in Table 5 demonstrate the broad applicability of larger batch sizes in fine-tuning, with the 4k batch size outperforming the 128 batch size across all but one metric (TruthfulQA).

## 3.5 EFFECT OF LEARNING RATE

We examined how different learning rates impact the model's downstream performance, using Granite 7B as a base model. We used the LAB hyperparameter configuration since it outperformed TULU. We conducted a learning rate sweep from $2 \times 10^{-5}$ to $1 \times 10^{-4}$. All other hyperparameters were kept constant to isolate the effect of the learning rate. We evaluated on MMLU, MTBench, Leaderboard (BBH, MuSR), ARC, and GSM8K benchmarks after the final phase of phased training. As shown in Table 6, the lowest learning rate of $2 \times 10^{-5}$ yielded the best performance on most benchmarks and comparable performance on the rest. As the learning rate increased, there was a consistent decline in benchmark performance. This trend suggests that lower learning rates enhance the model's ability to generalize to unseen tasks requiring knowledge, complex reasoning, and instruction following.

Table 6: Effect of Learning Rate Sweep on Benchmark Scores. Cells highlighted in green indicate better scores.

| Benchmark | Granite Base Pretrained | Learning Rates | | | |
|---|---|---|---|---|---|
| | | 2e-5 | 4e-5 | 8e-5 | 1e-4 |
| **MMLU** | 0.48 | 0.52 | 0.52 | 0.52 | 0.52 |
| **MTBench** | - | 6.76 | 6.64 | 6.53 | 6.47 |
| **Leaderboard (BBH)** | 0.09 | 0.10 | 0.09 | 0.09 | 0.08 |
| **Leaderboard (MuSR)** | 0.01 | 0.08 | 0.07 | 0.08 | 0.06 |
| **ARC** | 0.78 | 0.74 | 0.75 | 0.75 | 0.73 |
| **GSM8K** | 0.11 | 0.38 | 0.36 | 0.37 | 0.30 |

Lower learning rates may aid in retaining knowledge from previous training phases (e.g., instruction following and memorization) by preventing abrupt changes to the model's parameters. This is particularly important when fine-tuning on complex skills in Phase 10, as it requires the model to build upon its existing capabilities without forgetting prior knowledge. Lower learning rates also likely help prevent significant deviations from the pre-training parameters, thus enhancing generalization on benchmarks. Additionally, our experiments revealed that larger batch sizes did not require higher learning rates, as a lower learning rate of $2 \times 10^{-5}$ consistently provided better or comparable performance. Further details are provided in Appendix A.6.5.

## 3.6 EFFECT OF WARMUP STEPS

We investigated the impact of the number of warmup steps on the training process and final model performance. The warmup phase is traditionally considered crucial for stabilizing training, especially when using higher learning rates, by gradually increasing the learning rate from a small value to its target value over a specified number of steps (Goyal et al., 2017). We ran experiments with the Granite 7B model in the stacked setting using LAB hyperparameters—our best configuration—across three warmup setups: 0, 25, and 100 warmup steps, corresponding to approximately 0, 96,000, and 384,000 samples processed before reaching the target learning rate, respectively. As shown in Appendix A.6.6, the model trained without warmup steps achieved better performance on the MMLU benchmark and similar performance on MTBench compared to models trained with 25 or 100 warmup steps. The training curves for all configurations followed a similar trajectory, converging to comparable performance levels within approximately the same number of training steps, indicating that omitting warmup steps does not negatively affect the final model performance.

Although omitting the warmup simplifies the training process, it offers no advantage in terms of faster convergence. Furthermore, we monitored training dynamics such as gradient norms and loss values across the different warmup configurations. As shown in Appendix A.6.6, the gradient norms and loss values exhibited similar patterns across all configurations, indicating stable training even without a warmup phase.

## 3.7 EARLY TRAINING DYNAMICS AS PREDICTORS OF FINAL PERFORMANCE

We consistently observed that models exhibiting lower gradient norms and higher training loss values during training achieved better final performance on MMLU and MTBench. Figures 1 and 2 illustrate the correlation between early training dynamics—gradient norms and loss values—and final benchmark performances. The trend holds across different batch sizes, warmup steps, and learning rate schedules (see Appendix A.6.9 for additional results).

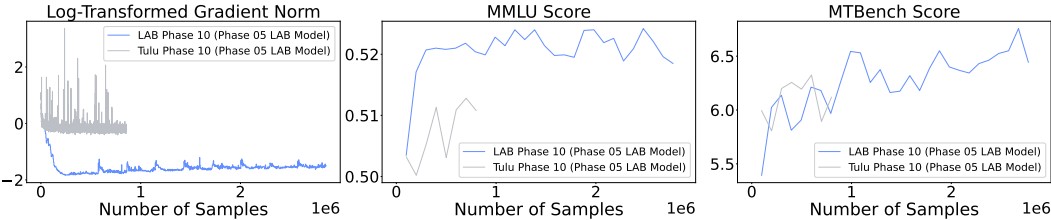

Figure 1: Correlation between early training dynamics and final performance on MMLU and MT-Bench benchmarks for TULU vs. LAB Phase 10 training.

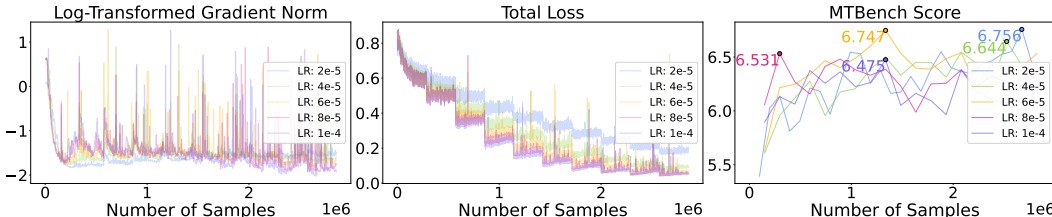

Figure 2: LAB Learning Rate (LR) Sweep: Training Dynamics and MTBench Performance. MMLU results are provided in Appendix A.6.9.

**TULU vs. LAB Phase 10 Training (Figure 1).** The LAB configuration achieved better final performance with lower gradient norms compared to the TULU configuration.

**LAB Learning Rate Sweep Experiments.** Models trained with a learning rate of $2 \times 10^{-5}$ demonstrated lower gradient norms initially, which increased toward the end of training, and higher loss throughout, ultimately resulting in superior final performance compared to models trained with higher learning rates. For the most effective learning rates, the gradient norm started at its lowest value and increased towards the end of training (Figure 2). Despite the higher gradient norms in the later stages, the associated loss remained higher throughout the entire training for these rates. This is consistent with the use of lower learning rates, which typically result in higher training loss but better generalization. Figure 2 shows the gradient norms and loss values for different learning rates, along with the final performance on MTBench. The lowest learning rates delivered superior results. Smaller learning rates may enable the model to stabilize the learning process initially and then gradually explore more challenging regions of the loss landscape as training progresses, leading to better generalization and final performance. We hypothesize that lower gradient norm values at the start of training contribute to a smoother and more stable optimization process, preventing the model from overfitting too quickly. This allows for gradual learning, which we hypothesize facilitates better exploration of the loss landscape as training progresses. The subsequent increase in gradient norm during later training stages may indicate that the model is delving into more complex regions of the parameter space, enhancing its ability to generalize. MMLU results are provided in Appendix A.6.9.

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

## A APPENDIX

### A.1 DISCUSSION, GUIDELINES FOR PRACTITIONERS, AND LIMITATIONS

**Balancing Performance and Efficiency.** Our results show a trade-off between performance and computational cost. Configurations such as higher batch sizes or lower learning rates achieve better final performance but take longer to converge. In contrast, hyperparameters yielding lower final performance often dominate early on before plateauing. For those with limited resources, smaller batch sizes or higher learning rates may be more efficient. For example, in stacked training, a 4k batch size outperforms 8k initially, and higher learning rates offer faster learning in early stages. Moreover, we encourage practitioners to monitor early training dynamics, such as gradient norms and loss values, as they correlate strongly with final model performance. Observing lower gradient norms and higher loss values during the initial phases of training can serve as reliable indicators of better generalization capabilities. This allows for early termination of suboptimal runs, conserving computational resources.

**Training Strategy Recommendations.** Based on our empirical evidence, we advocate for stacked over sequential phased training. This recommendation is supported by consistent performance gains and improved sample efficiency observed in Granite 3B, Granite 7B, and LLaMA 3B models. Stacked training simplifies the fine-tuning process and eliminates the need for phase-wise data management.

**Hyperparameter Selection.** We offer guidance on selecting batch sizes, learning rates, warmup steps, and learning rate schedules. Larger batch sizes (e.g., 4k and 8k) are recommended, as they have demonstrated superior performance across model sizes compared to smaller batch sizes like 128. Low learning rates are crucial for optimal performance. We found that $2 \times 10^{-5}$ works well for Granite models, while $1 \times 10^{-6}$ performs best for Mistral. Lower learning rates allow for more precise adjustments to the model weights, preventing overshooting in the optimization landscape. Practitioners should start with these values and, if necessary, perform a localized search by testing slightly higher or lower learning rates to find the optimal setting for their specific model. This approach significantly reduces the search space. Our experiments indicate that omitting warmup steps and using a constant learning rate instead of cosine decay does not negatively impact performance, simplifying the training process without sacrificing model quality.

**Limitations.** Our experiments focused on small (3B to 7B parameters) LLMs and were conducted on two model architectures: Granite (based on the Llama architecture) and Mistral. While our findings are promising, they may not directly generalize to larger models or other architectures. Future work should explore whether these observations hold for larger models and across a broader range of architectures, such as Gemma or others. Additionally, we did not investigate parameter-efficient fine-tuning strategies, such as LoRA, or explore how different pre-training objectives, tokenizer configurations, or optimizers (as we focused solely on Adam) might affect the applicability of our fine-tuning strategies. Furthermore, our evaluation centered on synthetic datasets generated from a comprehensive taxonomy covering various knowledge and skills, using benchmarks such as MMLU, MTBench, and LLM Leaderboard v2. We also explored the TULU dataset to understand fine-tuning across diverse datasets. However, confirming these findings across additional datasets and evaluation metrics would further strengthen the generalizability of our conclusions. Finally, we acknowledge that our experiments were conducted using a single seed due to computational constraints, which may introduce some noise into the observations.

### A.2 MODEL DETAILS

`Granite 3B` is composed of transformer layers (decoder blocks) that include multi-head self-attention mechanisms and feed-forward networks. It has a smaller hidden size and fewer attention heads, making it less computationally intensive and faster for both training and inference.

`Granite 7B` has more transformer layers and increased hidden dimensions, offers greater representational capacity. It also includes more attention heads, enabling it to capture more complex language patterns and long-range dependencies.

`Llama 3.2 3B` employs a scaled-down transformer architecture with fewer layers and a reduced hidden size compared to larger models in the Llama family. It maintains the core design principles of

its larger counterparts, including rotary positional embeddings and optimized attention mechanisms, while balancing performance and efficiency for resource-constrained environments.

`Mistral 7B` utilizes advanced attention mechanisms, including multi-query attention and Sliding Window Attention, which enhance efficiency and reduce memory usage during inference. With 32 layers and 32 attention heads, it is designed for improved performance on benchmarks, particularly in logical reasoning and commonsense tasks, while maintaining competitive resource demands.

## A.3  DATASETS DETAILS

The datasets were curated using a taxonomy-driven approach to ensure comprehensive coverage of instruction-following, foundational knowledge, and compositional skills. The taxonomy hierarchically organizes tasks into three main branches—knowledge, foundational skills, and compositional skills—each further divided into granular subcategories. For each subcategory, manually written instruction-response pairs served as seed examples. These examples guided synthetic data generation using teacher models (e.g., Mixtral-7x8B) to expand the dataset while maintaining high quality and diversity. For knowledge data, reliable sources such as textbooks and technical manuals provided a grounding for synthetic questions and responses. Foundational skills data were drawn from public datasets covering essential areas like mathematics, coding, and reasoning. Compositional skills were synthesized using a taxonomy-guided approach to combine knowledge and foundational skills for complex tasks, such as writing detailed emails or generating logical arguments. We provide details about the datasets we used in Table 7.

Table 7: Summary of datasets used in different phases.

| Phase | Description | # Samples |
|---|---|---|
| **Phase 00** | Instruction following warmup: simple, template-based instruction-response pairs to transition the base models to instruction-following behavior. | 308343 |
| **Phase 05** | Foundational knowledge acquisition: synthetically generated question-answer pairs from textbooks covering a wide range of disciplines up to graduate-level courses. | 231178 |
| **Phase 10** | Complex skills development: synthetic data generated using a taxonomy of skills, including tasks like poetry, email writing, logical reasoning, coding, and more. | 285966 |
| **All-Phases** | Combination of phases 00, 05, and 10, exposing models to all data types simultaneously. | 825487 |

## A.4  TRAINING INFRASTRUCTURE AND OPTIMIZATION

To handle large batch sizes and optimize computational efficiency, we use an optimized training infrastructure.

**Optimizer.** Across all experiments, we use the Adam optimizer with $\beta_1 = 0.9$ and $\beta_2 = 0.95$. By adjusting $\beta_2 = 0.95$, we reduce the emphasis on the variance of past gradients, which is beneficial when training with large batch sizes that provide more stable gradient estimates.

**Batching and Gradient Accumulation.** To achieve the large effective batch sizes required for our experiments, we employed gradient accumulation techniques. Gradient accumulation involves accumulating gradients over multiple forward and backward passes before taking a gradient step (i.e., updating the model weights). This effectively increases the batch size without necessitating additional memory to store larger batches in a single pass. For instance, in a single-node setup with 8 GPUs, we set a micro-batch size per GPU and used gradient accumulation steps to reach an effective batch size of 3,840 samples. Specifically, if each GPU processes a micro-batch of $b$ samples and we accumulate gradients over $k$ steps, the effective batch size $B$ is $B = b \times k \times N$, where $N$ is the number of GPUs. In multi-node setups with 64 GPUs, we could process the entire batch in a single step without accumulation due to the distributed computational resources. This approach allowed us to simulate very large batch sizes, to investigate their impact on model performance.

**Efficient Distributed Sampling.** We implement a variant of Multipack distributed sampler (MultipackSampler, 2024), which offers significant advantages over naive sampling approaches in distributed training of LLMs. Drawing on concepts from the identical machine scheduling problem (Graham, 1966), our implementation uses an approximate solution at the sample level, achieving near-optimal GPU utilization. Our variant extends the original design by accounting for padding, crucial for non-linear attention mechanisms like scaled dot-product attention (Vaswani et al., 2017), and clustering together samples of similar length. It ensures that even with padding, no GPU exceeds a pre-determined token capacity, which we calculate to maintain an expected micro-batch size that satisfies:

$$\text{E[Effective Batch Size]} = \text{E[Micro Batch Size} \times \text{Gradient Accumulation Steps]}$$

This approach balances computational load across GPUs, resulting in improved training throughput and stability. Additionally, our sampler supports both linear attention mechanisms, such as FlashAttention (Dao et al., 2022), and traditional non-linear attention, making it versatile for various model architectures.

### A.5 Experimental Design

We investigate how training strategies, batch sizes, learning rates, and warmup steps influence LLM fine-tuning. We systematically vary these factors while holding other parameters constant to isolate their individual effects.

**Impact of Batch Size and Training Strategies.** We examine how different batch sizes influence model performance and training dynamics in both stacked and phased training settings. Our hypothesis is that larger batch sizes will improve model performance in stacked training by ensuring sufficient data diversity within each batch, allowing for more robust gradient updates. In contrast, their impact on phased training may be less pronounced. On the other hand, this improvement may come at the cost of reduced sample efficiency. While larger batch sizes may achieve better final performance, they typically require more training samples and computational resources due to the higher number of samples used per gradient step. Conversely, smaller batch sizes could achieve comparable performance with fewer samples, especially in phased training, where each batch is inherently constrained to a specific data type, limiting intra-batch diversity. We formalize these hypotheses below and explore the trade-offs between performance gains and computational efficiency in our experiments.

- **Hypothesis 1.** Stacked training may underperform at smaller batch sizes due to insufficient diversity within each batch. A smaller batch may not capture the wide range of data types present in the combined dataset, leading to less effective learning. In contrast, larger batch sizes in stacked training could match or surpass phased training by capturing a wider range of signals in each gradient update.

- **Hypothesis 2.** While the stacked approach simplifies the training pipeline by eliminating the need for phase selection of data, it can be less sample efficient. Learning all types of data simultaneously could require more steps for the model to adequately learn the complex and diverse patterns in the combined dataset. This translates to worse sample efficiency, as the model may need more gradient updates to converge.

**Learning Rate Exploration.** We conduct a learning rate sweep to examine its influence on training dynamics and final model performance. We explore whether larger batch sizes require higher learning rates, hypothesizing that increased gradient stability at higher batch sizes may allow for more aggressive learning rate schedules without causing instability. Additionally, we hypothesize that lower learning rates (e.g., $2e-5$) may help generalization by helping the model stay closer to the pre-trained parameters, preventing overfitting on the fine-tuning data and forgetting of previously learned information. Additionally, we evaluate the effects of different learning rate schedules (constant vs. cosine decay) on model performance, as cosine decay is widely recognized for its smooth convergence properties. Our goal is to empirically determine whether this schedule offers tangible benefits in our specific setting.

**Warmup Steps Analysis.** Warmup steps are commonly used in fine-tuning LLMs to stabilize training. We investigate whether reducing or removing warmup steps can accelerate convergence without sacrificing final model performance.

**Training Dynamics and Early Performance Indicators.** We monitor key training dynamics, such as gradient norms and loss values, to explore their correlation with the model's final performance metrics on benchmarks (MMLU and MTBench). Monitoring gradient norm and loss during training provides insights into the smoothness of the optimization trajectory, with lower gradient norms suggesting traversal through flatter regions of the loss landscape, which, as discussed later, can influence final model performance. The goal was to investigate whether early-stage indicators, such as a lower gradient norm and higher loss values during the initial phase of training or consequently throughout the entire training, can serve as reliable predictors of better performance on benchmarks. This approach could allow for the early termination of suboptimal runs, optimizing computational resources by focusing only on models that demonstrate promising training dynamics. By closely examining these metrics across multiple learning rate configurations, batch sizes, and training strategies, we aimed to understand how these dynamics reflect the underlying optimization process and its relationship to final task performance, without the need for full training to completion. Identifying these early indicators is critical for advancing sample-efficient training methodologies, especially in large-scale experiments.

**Adaptations for Different Model Architectures and Sizes.** To evaluate the generalizability of our findings across different model architectures and model sizes, we extended our experiments to include the Granite 3B, Mistral 7B, and LLaMA 3B models. Mistral models incorporate architectural optimizations and differ from the Granite models in aspects such as tokenization and layer configurations. Similarly, the LLaMA 3B model, while sharing foundational similarities with Granite, is recognized for its efficient scaling laws and pretraining strategies. We evaluated both training strategies (stacked vs. phased) and adapted specific training hyperparameters (e.g., batch size, learning rate, warmup steps) to verify the robustness of our methodology and ensure our results are applicable to a diverse range of small-sized LLM architectures.

## A.6 ADDITIONAL RESULTS

### A.6.1 STACKED TRAINING VS. SEQUENTIAL PHASED TRAINING

Contrary to our initial hypothesis that stacked training might underperform at smaller batch sizes due to insufficient gradient stability, our results show that stacked training achieves better or comparable performance to phased training consistently at both 128 and 4,000 batch sizes. The performance comparison across MTBench and MMLU benchmarks indicates that stacked training slightly outperforms phased training at each batch size, suggesting that batch size does not significantly impact the difference between the two training strategies. Instead, stacked training's exposure to the entire dataset in each epoch, even at smaller batch sizes, may help maintain stability in learning, effectively supporting generalization across diverse types of data without requiring phased partitioning.

In Figure 3, we compare the performance of both training strategies using the LAB hyperparameter configuration, which provided the best overall results for both approaches. Figure 3a shows the final MTBench performance, where stacked training outperformed phased training by 0.01 points. Figure 3b illustrates that stacked training is also more sample-efficient, with the best performance points annotated by the number of samples required to reach them. Note that the line for phased training begins partway through, as samples from Phases 00 and 05 were already included. This applies consistently to all similar figures presented in this paper.

In addition to the MTBench results, we include the MMLU performance comparisons here. MMLU is split into two plots for clarity and readability. Figure 4 shows the final MMLU performance using LAB hyperparameters for both stacked and phased training strategies. Stacked training outperformed phased training on the MMLU benchmark by 0.01 points, consistent with the observations from MTBench.

Figure 5 illustrates the sample efficiency comparison for MMLU. Similar to the MTBench results, stacked training achieves higher MMLU performance more quickly than phased training. These results reinforce our findings that stacked training not only improves performance but also enhances sample efficiency on both MMLU and MTBench benchmarks.

To investigate whether phased training might be effective when phases are split based on difficulty, we conducted an additional experiment. In this setup, we partitioned the dataset into two phases based on difficulty, using the length of free-form answers as a proxy for difficulty.

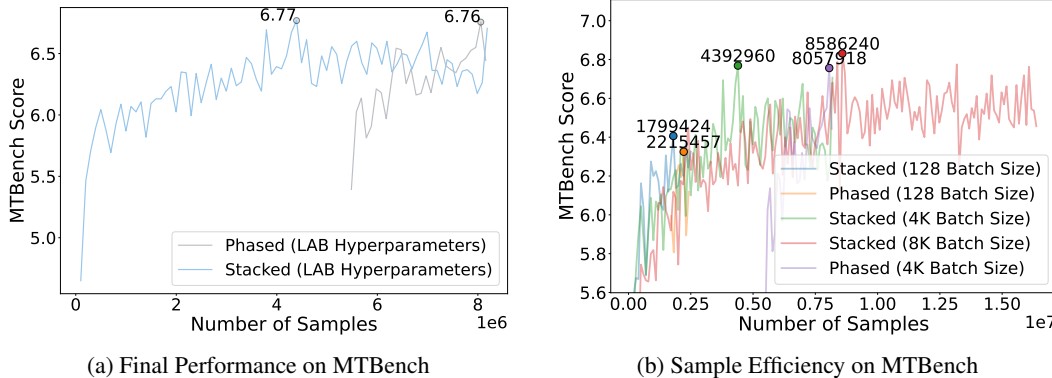

Figure 3: Comparison of stacked and phased training strategies on MTBench using LAB hyperparameters.

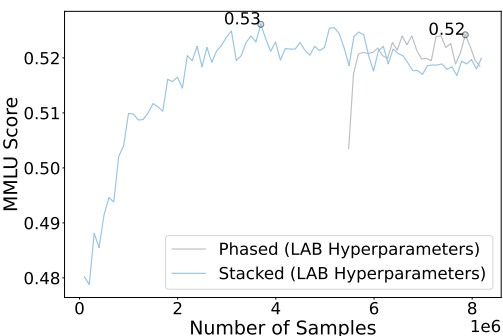

Figure 4: Final MMLU Performance comparison using LAB hyperparameters: stacked vs. phased training.

- **Phase I:** The bottom 50% of the data containing short sentences.

- **Phase II:** The top 50% of the data containing long sentences, plus a 1% subset of the short sentences as a replay buffer when transitioning to long sentences.

We fine-tuned the Granite 7B base model using the same hyperparameters—a batch size of 4,000 and a learning rate of $2 \times 10^{-5}$—under both the phased and stacked training strategies. Our results (Figure 6) showed no significant difference between phased and stacked training in this setting. Both performed similarly, with stacked training slightly outperforming phased training across all benchmarks. This suggests that even when the data is carefully partitioned based on difficulty, phased training does not improve model performance over stacked training. Moreover, phased training remains less sample-efficient due to the additional time and samples required to determine the optimal checkpoint for phase transitions.

### A.6.2 IMPACT OF BATCH SIZE

Figure 8 shows the performance of different batch sizes in stacked and phased training on the MT-Bench benchmark. Similarly, Figure 7 highlights the impact of batch size on model performance in both stacked and phased training on the MMLU benchmark. These results consistently demonstrate that larger batch sizes lead to better final performance. While smaller batch sizes initially reach higher performance levels more quickly, they plateau earlier, allowing larger batch sizes to surpass them with extended training.

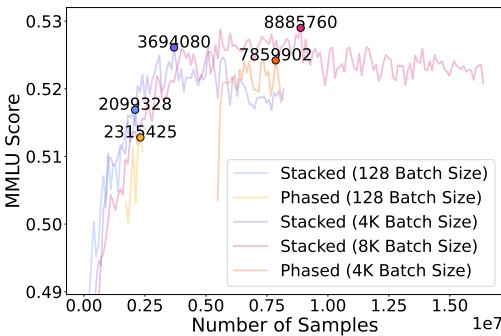

Figure 5: MMLU Sample efficiency comparison between stacked and phased training.

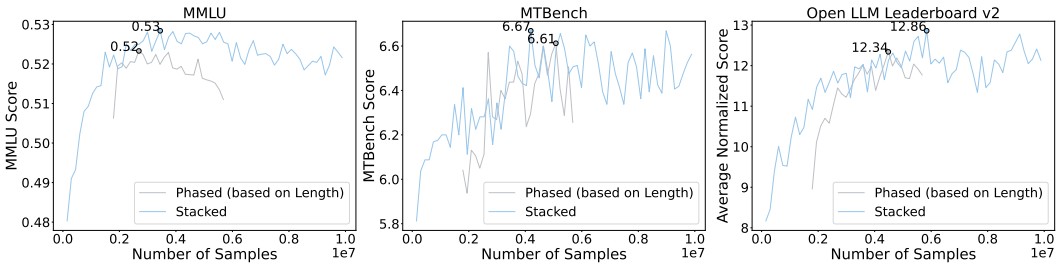

Figure 6: Performance comparison of stacked vs. phased training on difficulty-partitioned data (by answer length) across comprehensive benchmarks.

### A.6.3 EFFECT OF LEARNING RATE SCHEDULES ON LARGE BATCH SIZES

As shown in Figure 9, the models trained with a constant learning rate (no decay) performed on par with those trained with a cosine decay schedule on both MMLU and MTBench, and in some cases even outperformed them, particularly on MTBench.

### A.6.4 TULU VS. LAB

**Memorization and Generalization.** We focused on Phase 05 in the sequential phased training strategy, which is designed to augment the model's foundational knowledge and memorization of facts. We evaluated the models using specific MMLU subjects related to memorization, including history, law, and science domains. We compared the performance of models starting from both the base Granite model and the best checkpoint obtained from Phase 00 training. Table 8 shows that models trained with LAB hyperparameters outperform those trained with TULU hyperparameters on the memorization-focused MMLU tasks. We evaluated the models' generalization abilities after Phase 10 in the sequential phased training strategy, which focuses on complex skills and compositional tasks. The MTBench benchmark was used to assess performance on tasks requiring reasoning, problem-solving, and adaptation to unseen scenarios. As shown in Table 8, the model trained with LAB hyperparameters significantly outperforms the one trained with TULU hyperparameters on MTBench.

To ensure a fair comparison, we ran each experiment for the same number of gradient steps, resulting in different number of samples due to different batch sizes, as seen in Figure 10. Figure 10a shows that models trained with LAB hyperparameters outperform those trained with TULU hyperparameters on the memorization-focused MMLU tasks. Additionally, as shown in Figure 10b, the model trained with LAB hyperparameters significantly outperforms the one trained with TULU hyperparameters on MTBench. Table 9 shows that LAB performs better than TULU across all benchmarks.

### A.6.5 EFFECT OF LEARNING RATE

As shown in Figure 11, the lowest learning rate of $2 \times 10^{-5}$ yielded the best performance on the MTBench Benchmark.

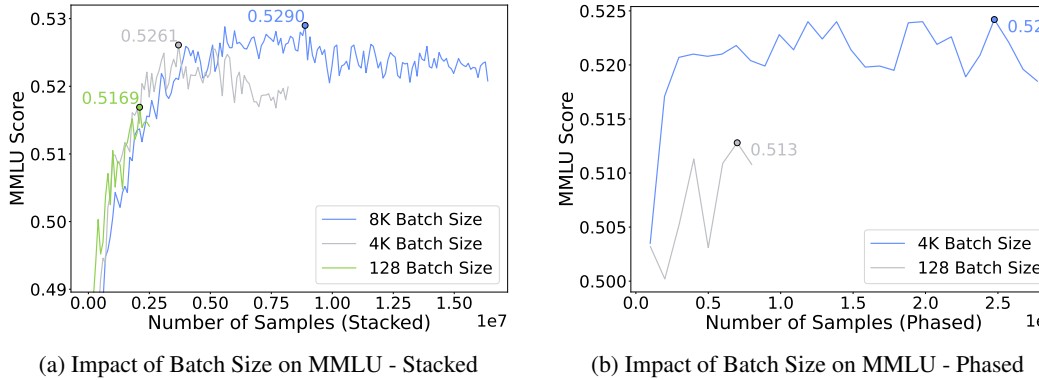

(a) Impact of Batch Size on MMLU - Stacked          (b) Impact of Batch Size on MMLU - Phased

Figure 7: Impact of batch size on model performance in stacked and phased training on MMLU benchmark.

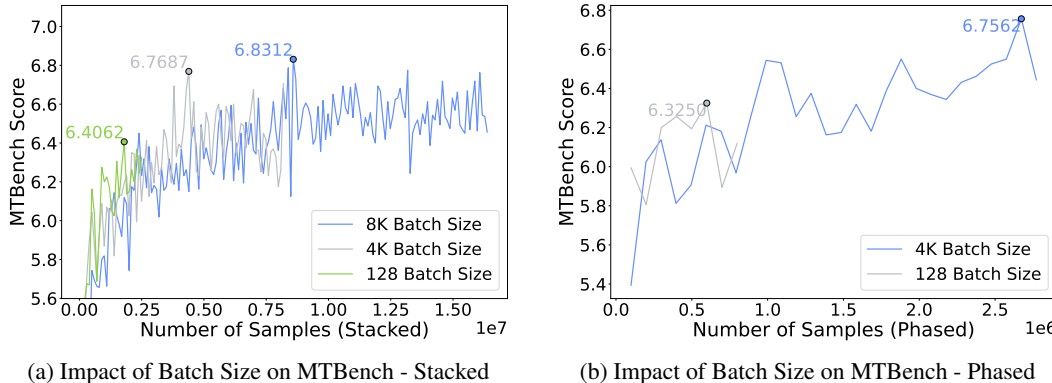

(a) Impact of Batch Size on MTBench - Stacked          (b) Impact of Batch Size on MTBench - Phased

Figure 8: Impact of batch size on model performance in stacked and phased training on MTBench benchmark.

Table 8: Comparison of TULU vs. LAB on MMLU Memorization and MTBench Generalization Scores. Cells highlighted in green indicate better scores, and blue indicates higher sample efficiency (fewer samples used).

| Benchmark | Model | Score | Samples |
|---|---|---|---|
| **MMLU (Memorization)** | Granite Base | 0.48 | - |
| | TULU (Base) | 0.59 | 199,936 |
| | LAB (Base) | 0.61 | 1,597,440 |
| | TULU (Phase 00) | 0.60 | 599,808 |
| | LAB (Phase 00) | 0.62 | 1,098,240 |
| **MTBench (Generalization)** | TULU | 6.33 | 599,808 |
| | LAB | 6.76 | 2,673,216 |

We investigated whether larger batch sizes necessitate higher learning rates, based on the premise that with a larger batch size, the model processes more samples before each gradient step, potentially benefiting from a higher learning rate to make more significant updates and to maintain the variance of the gradient when compared to smaller batch sizes. Additionally, since larger batches result in fewer gradient steps over the same number of epochs, increasing the learning rate might improve training efficiency.

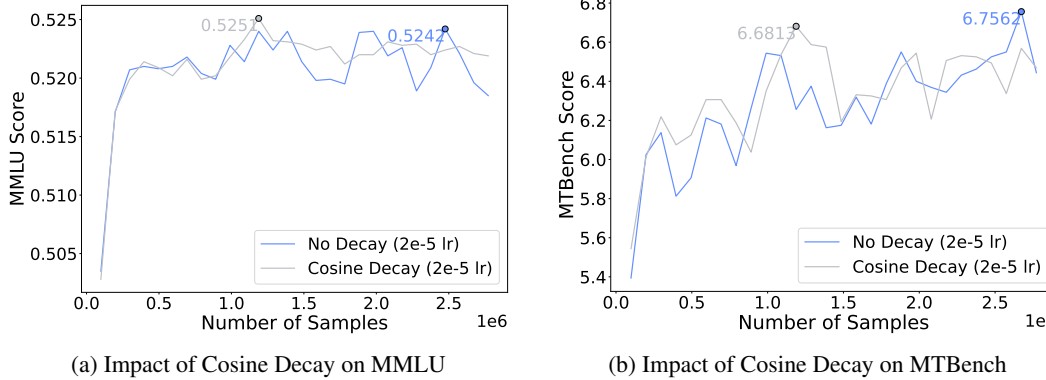

(a) Impact of Cosine Decay on MMLU

(b) Impact of Cosine Decay on MTBench

Figure 9: Comparison of learning rate schedules with a batch size of 3,840 samples on MMLU and MTBench benchmarks.

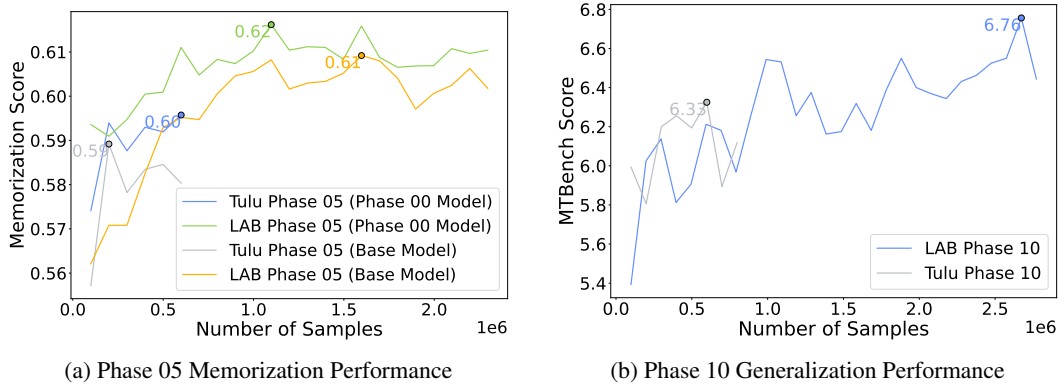

(a) Phase 05 Memorization Performance

(b) Phase 10 Generalization Performance

Figure 10: Comparison of TULU vs. LAB on memorization and generalization.

Table 9: Comparison of LAB vs. TULU Hyperparameter Configurations on Various Benchmarks. Cells highlighted in green indicate better scores, and blue indicates higher sample efficiency (fewer samples used).

| Benchmark | Score | | | Samples | |
|---|---|---|---|---|---|
| | **Granite Base** | **LAB** | **TULU** | **LAB** | **TULU** |
| **Leaderboard (BBH)** | 0.09 | 0.10 | 0.08 | 8,057,918 | 599,808 |
| **Leaderboard (MuSR)** | 0.01 | 0.07 | 0.03 | 8,057,918 | 599,808 |
| **ARC** | 0.78 | 0.75 | 0.74 | 8,057,918 | 599,808 |
| **GSM8K** | 0.11 | 0.37 | 0.36 | 8,057,918 | 599,808 |

Our experiments compared models trained with different learning rates across batch sizes of 128, 3,840, and 7,680 samples. The runs included TULU hyperparameters at learning rates of $2 \times 10^{-5}$ and $3 \times 10^{-5}$, and LAB hyperparameters with learning rates ranging from $2 \times 10^{-5}$ to $1 \times 10^{-4}$. As shown in Figure 12, regardless of batch size, the lower learning rate of $2 \times 10^{-5}$ consistently resulted in better or comparable performance on both MMLU and MTBench benchmarks. For instance, with a batch size of 128, performances were similar for both the learning rates. For larger batch sizes of 3,840 and 7,680, the $2 \times 10^{-5}$ learning rate performed on par or better than higher learning rates.

A possible explanation for our findings is that large batches yield more stable gradient estimates by averaging over more samples, which allows effective progress at lower learning rates without risking instability. Higher learning rates with large batches, however, can cause the model to take

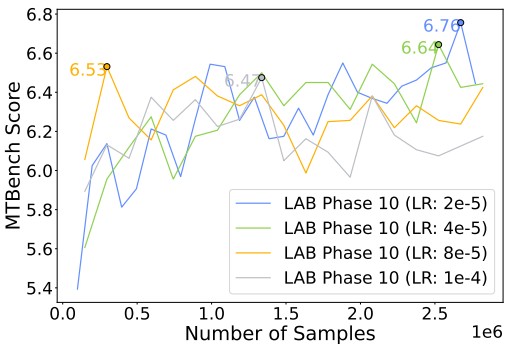

Figure 11: MTBench performance after Phase 10 training with different learning rates.

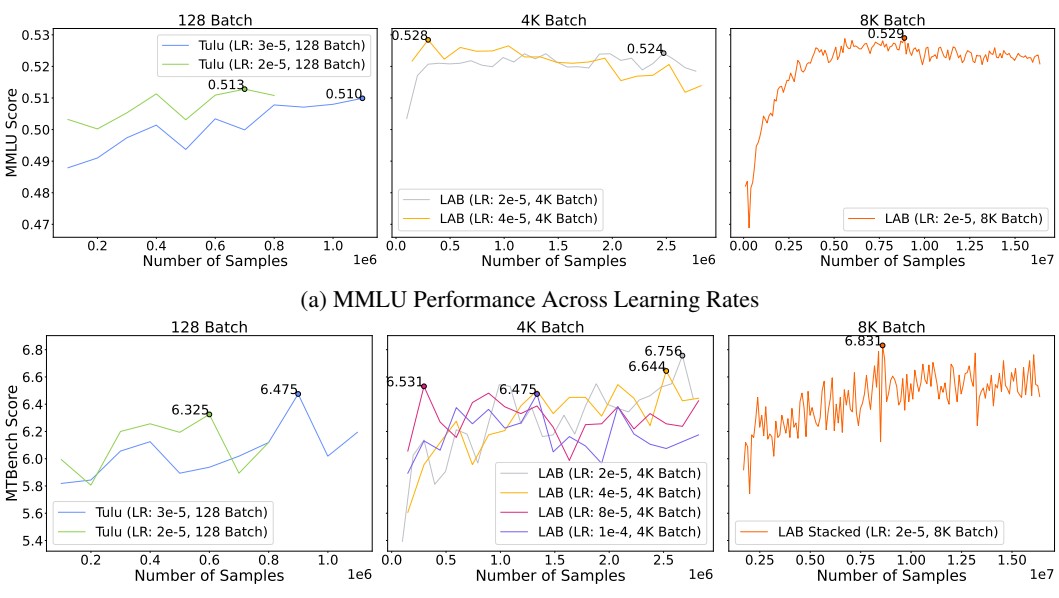

(a) MMLU Performance Across Learning Rates

(b) MTBench Performance Across Learning Rates

Figure 12: Performance comparison across different learning rates and batch sizes on MMLU and MTBench benchmarks.

larger steps that risk moving too far (Hoffer et al., 2017) from the pre-trained parameters, potentially overshooting the minima.

### A.6.6 EFFECT OF WARMUP STEPS

Figure 13 shows the performance comparison with different warmup steps: 0, 25, and 100 warmup steps, on the MMLU and MTBench benchmarks, respectively. The model trained without warmup steps achieved better performance on MMLU and similar performance on MTBench, suggesting that warmup steps may not be essential for stable and effective training.

### A.6.7 ADAPTATION TO A DOMAIN-SPECIFIC DATASET

To evaluate the generalizability of our findings to domain-specific datasets, we conducted experiments using a Math, Reasoning, and Code (MRC) dataset. This dataset specializes in mathematical problem-solving, logical reasoning, and programming tasks, representing a focused domain compared to our original diverse dataset.

We evaluated the models on several benchmarks, including GSM8K (Cobbe et al., 2021), ARC (Clark et al., 2018), and the Open LLM Leaderboard v2 benchmarks including MATH and MuSR.

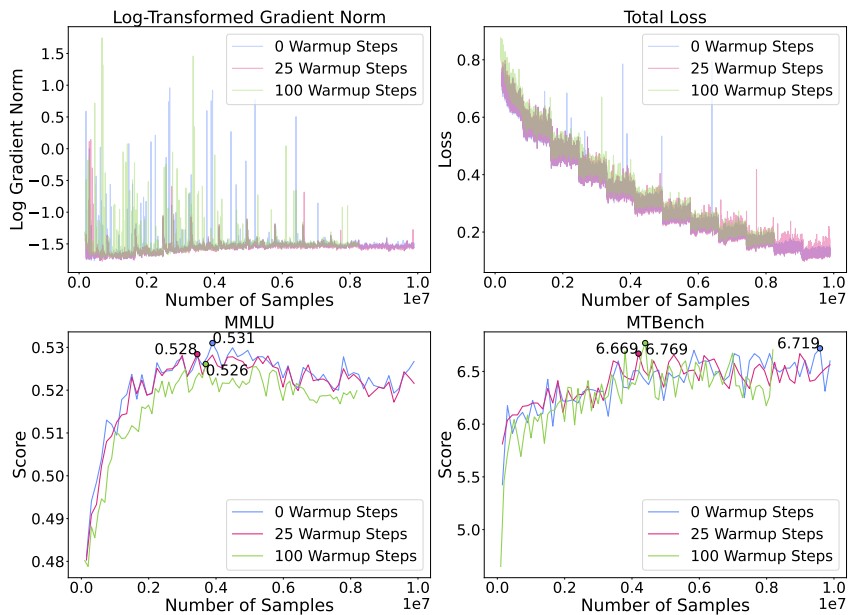

Figure 13: MMLU and MTBench performance of the Granite 7B LAB model with varying warmup steps: 0, 25, and 100 steps.

We compare the LAB and TULU hyperparameter configurations on the MRC dataset. Using the LLaMA 3B model, we fine-tuned with both configurations: LAB used a batch size of 4,000 and a constant learning rate, while TULU used a batch size of 128 with warmup and linear decay. As shown in Table 10, the LAB configuration outperforms TULU across all evaluation metrics, reaffirming that larger batch sizes and simplified learning rate schedules are beneficial even when fine-tuning on domain-specific data.

Table 10: Comparison of LAB vs. TULU Hyperparameter Configurations on the MRC Dataset Using the LLaMA 3B Model. Cells highlighted in green indicate better scores, and blue indicates higher sample efficiency (fewer samples used).

| Benchmark | Score | | | Samples | |
|---|---|---|---|---|---|
| | LLaMA Base | LAB | TULU | LAB | TULU |
| **Leaderboard (MATH Lvl 5)** | 0.02 | 0.04 | 0.04 | 9,980,259 | 3,468,664 |
| **Leaderboard (MuSR)** | 0.05 | 0.08 | 0.04 | 16,966,128 | 2,973,753 |
| **ARC** | 0.78 | 0.75 | 0.68 | 2,745,290 | 247,372 |
| **GSM8K** | 0.27 | 0.69 | 0.66 | 12,225,143 | 5,450,009 |

Additionally, we fine-tuned the LLaMA 3B model using both the stacked and sequential phased training strategies with LAB hyperparameters. For phased training, as described in Appendix A.6.1, the dataset was partitioned into two phases based on response length: Phase I with shorter responses (bottom 50%) and Phase II with longer responses (top 50%). As shown in Table 11, stacked training demonstrates slightly higher performance and greater sample efficiency compared to phased training across all benchmarks.

These findings demonstrate that our recommendations regarding training strategies and hyperparameters generalize to domain-specific datasets, supporting their applicability in specialized fine-tuning scenarios.

Table 11: Comparison of Stacked vs. Phased Training Strategies on the MRC Dataset Using the LLaMA 3B Model. Cells highlighted in green indicate better scores, and blue indicates higher sample efficiency (fewer samples used).

| Benchmark | Score | | | Samples | |
|---|---|---|---|---|---|
| | LLaMA Base | Stacked | Phased | Stacked | Phased |
| Leaderboard (MATH Lvl 5) | 0.02 | 0.04 | 0.03 | 9,980,259 | 18,455,850 |
| Leaderboard (MuSR) | 0.05 | 0.08 | 0.08 | 16,966,128 | 18,206,922 |
| ARC | 0.78 | 0.75 | 0.71 | 2,745,290 | 14,964,367 |
| GSM8K | 0.27 | 0.69 | 0.67 | 12,225,143 | 14,964,367 |

### A.6.8 ADAPTATIONS TO DIFFERENT MODEL SIZES AND ARCHITECTURES

To assess the scalability and generality of our findings, we extended our experiments to different model families, architectures, and sizes, specifically testing the Mistral 7B model, the Granite 3B model, and the LLaMA 3B model.

**Adaptation to a New Architecture.** We performed stacked training experiments with the Mistral 7B model, varying batch sizes (128 and 3,840) and learning rates ($1 \times 10^{-6}$, $5 \times 10^{-6}$, and $2 \times 10^{-5}$). We report downstream benchmark scores in MTBench and LLM Leaderboard v2, which includes MMLU-Pro, an enhanced version of MMLU that integrates more challenging, reasoning-focused questions and expands the choice set to better differentiate model capabilities (Wang et al., 2024). Our findings, illustrated in Figure 14, indicate that higher batch sizes lead to improved performance on MTBench and equivalent performance on Leaderboard benchmarks. Specifically, a batch size of 4k combined with a learning rate of $1 \times 10^{-6}$ yields the best results, as higher batch sizes and lower learning rates have a stabilizing effect on training, offering similar advantages by reducing noise/size of updates. Conversely, increasing the learning rate or reducing the batch size (e.g., using learning rates above $1 \times 10^{-6}$ with a 4k batch size, as shown in Figure 15, or a batch size of 128 with a learning rate of $1 \times 10^{-6}$) negatively impacts downstream performance.

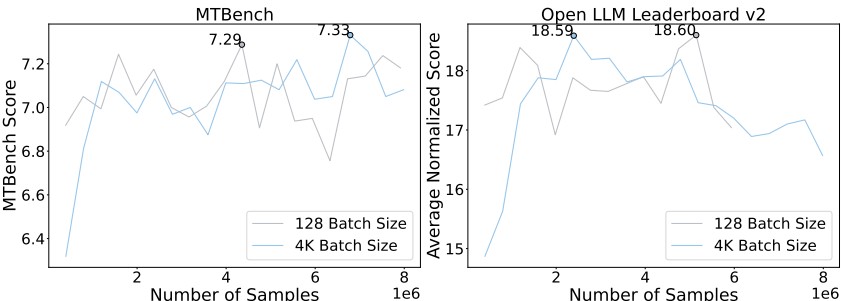

Figure 14: Benchmark performance comparison of different batch sizes for the Mistral 7B model.

While previous studies suggest higher learning rates are beneficial with larger batch sizes during training from scratch (Smith, 2017; Goyal et al., 2017), our findings indicate that, for fine-tuning pre-trained models, lower learning rates are preferable to minimize forgetting and maintain downstream performance. We reason that this discrepancy arises because, starting from a pre-trained model at a local minimum in the loss landscape, we aim to avoid moving too far from that minimum during fine-tuning to prevent forgetting what was learned during pre-training. Larger batch sizes and lower learning rates reduce stochasticity in the optimization process, leading to smaller, more stable updates that help the model stay closer to the pre-trained parameters while effectively adapting to new data. This aligns with findings from (Hoffer et al., 2017) that smaller batch sizes lead weights further from initialization due to higher estimation noise, while larger batch sizes keep weights closer to initialization by reducing the diffusion rate in the weight space. Therefore, using larger batch sizes and/or lower learning rates helps preserve the pre-trained knowledge while allowing the model to adapt to new tasks.

We conducted a learning rate sweep for the Mistral 7B model to determine which learning rate yields the best final performance. Our objective was to apply the methodology used for finding the optimal learning rate with the Granite models to the Mistral architecture. This methodology involves starting with a low learning rate. A low learning rate helps prevent large, destabilizing weight updates, allowing the model to fine-tune its parameters gradually and avoid overfitting. Additionally, lower learning rates facilitate more precise adjustments to the model weights, which is particularly important when adapting pre-trained models to new tasks or domains without forgetting previously learned information.

Our proposed methodology for identifying optimal hyperparameters involves starting with a baseline and iteratively testing slightly higher and lower values to detect performance improvements. For example, with learning rate, we began the search at $2 \times 10^{-5}$ (effective for Granite) and adjusted incrementally to refine the optimal range based on empirical results. This approach serves as a general prescription for all hyperparameters, allowing systematic fine-tuning. Using this method, we identified $1 \times 10^{-6}$ as the optimal learning rate for Mistral among the learning rates we tested.

The results are presented in Figure 15. We check if what we have observed before for Granite—that is, the general trend where lower gradient norms and higher loss are associated with better generalization and final performance—also applies to Mistral. Specifically, the lowest learning rate, $1 \times 10^{-6}$, produced the best overall performance on the MMLU benchmark. An interesting pattern emerged, similar to that observed with the Granite model: for the most effective learning rates, the gradient norm started at its lowest value and increased towards the end of training. Despite the higher gradient norm in the later stages, the associated loss remained higher throughout training (which is expected because lower learning rates typically result in higher loss values during training). This suggests that higher loss values may be an indicator of better model generalization. These observations confirm that the correlation between early training dynamics and final downstream performance is consistent across different model architectures.

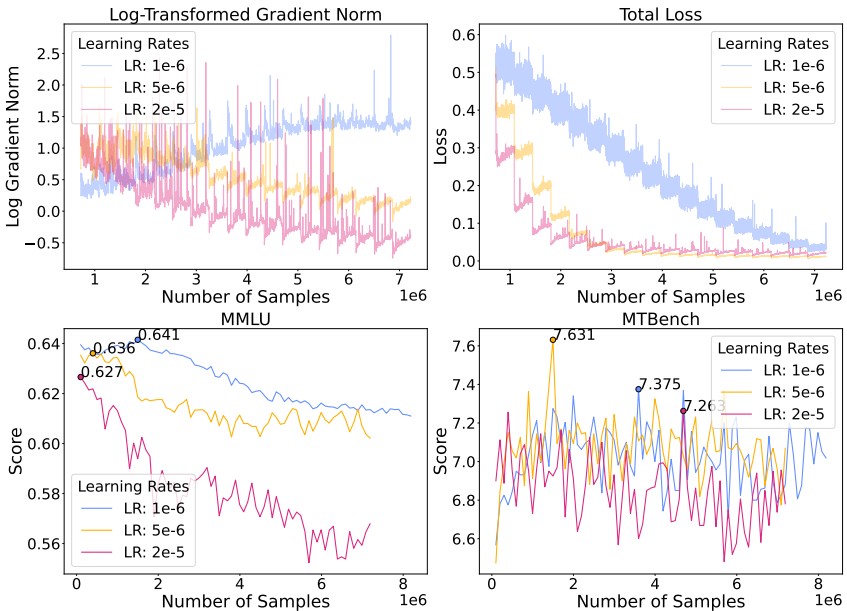

Figure 15: Training dynamics for Mistral 7B with different learning rates, and their final performance on MMLU and MTBench benchmarks.

**Adaptation to Different Model Sizes.** We also examined whether our findings hold for smaller models by conducting experiments with the Granite 3B model. Specifically, we compared an 8k batch size with stacked training versus a 4k batch size with phased training. Our goal was to determine if the observations regarding batch size and training strategies for the Granite 7B model extend to the 3B model. In the 8k stacked setting for the Granite 3B model, we observed a lower gradient norm, higher loss, and improved MMLU performance compared to the 4k phased configuration. This trend is illustrated in Figure 16.

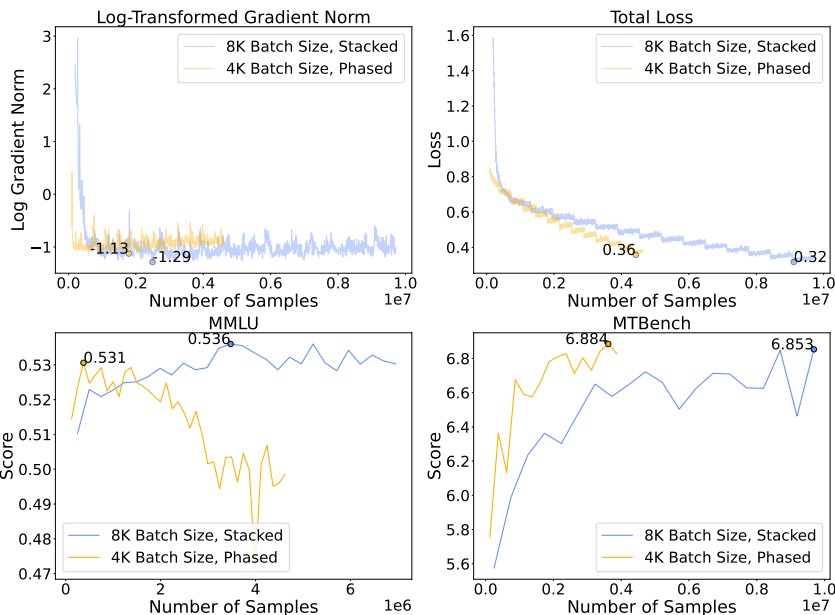

Figure 16: Training dynamics for Granite 3B with different batch sizes and training strategies (8k stacked vs. 4k phased), and their final performance on MMLU and MTBench benchmarks.

The larger batch size likely improves performance by increasing data diversity within each batch, covering a range of tasks, skills, and knowledge. This diversity reduces gradient variance, promoting stable updates and helping the model retain pre-trained knowledge without significant forgetting. The lower gradient norm in the 8k stacked setting suggests that the model is settling into a flatter, more generalizable region of the loss landscape, while the higher loss indicates reduced risk of overfitting by maintaining a broader exploration. Together, these factors likely contribute to the superior performance of the 8k stacked configuration on the MMLU benchmark. These results suggest that the correlation between early training dynamics and final performance holds across different model sizes.

**Generalization to a Different Model Family and Size.** To assess whether our findings extend to a different model architecture and size at the same time, we conducted experiments using the LLaMA 3B model (Touvron et al., 2023). We note that the Granite model shares the same architecture as the LLaMA model. Hence we believe that the findings in this paper can generalize across the LLaMA model family. We fine-tuned the model using both stacked and phased training strategies, as well as comparing the LAB and TULU hyperparameter configurations.

Table 12: Comparison of Stacked vs. Phased Training Strategies Using the LLaMA 3B Model. Cells highlighted in green indicate better scores, and blue indicates higher sample efficiency (fewer samples used).

| Benchmark | Score | | | Samples | |
|---|---|---|---|---|---|
| | **LLaMA Base** | **Stacked** | **Phased** | **Stacked** | **Phased** |
| **Leaderboard (BBH)** | 0.14 | 0.19 | 0.18 | 7,734,723 | 6,734,847 |
| **Leaderboard (MATH Lvl 5)** | 0.01 | 0.02 | 0.01 | 250,089 | 4,490,320 |
| **Leaderboard (MuSR)** | 0.05 | 0.22 | 0.11 | 10,979,309 | 4,988,983 |
| **MMLU** | 0.56 | 0.57 | 0.53 | 6,986,437 | 5,737,613 |
| **ARC** | 0.78 | 0.78 | 0.75 | 2,744,559 | 7,483,283 |
| **GSM8K** | 0.27 | 0.51 | 0.45 | 3,742,399 | 6,734,847 |
| **MTBench** | - | 5.00 | 4.30 | 9,232,227 | 6,734,847 |

For phased training, as described in Appendix A.6.1, the dataset was partitioned into two phases based on response length: Phase I with shorter responses (bottom 50%) and Phase II with longer responses (top 50%). We compared the LAB configuration (batch size of 4,000 with constant learning rate) to the TULU configuration (batch size of 128 with warmup and linear decay). The models were evaluated on benchmarks including MMLU, MTBench, GSM8K, ARC, and the Open LLM Leaderboard v2 benchmarks including BBH, MATH, and MuSR.

Table 13: Comparison of LAB vs. TULU Hyperparameter Configurations on the LLaMA 3B Model. Cells highlighted in green indicate better scores, and blue indicates higher sample efficiency (fewer samples used).

| Benchmark | Score | | | Samples | |
|---|---|---|---|---|---|
| | **LLaMA Base** | **LAB** | **TULU** | **LAB** | **TULU** |
| **Leaderboard (BBH)** | 0.14 | 0.19 | 0.17 | 7,734,723 | 2,473,477 |
| **Leaderboard (MATH Lvl 5)** | 0.01 | 0.02 | 0.01 | 250,089 | 741,924 |
| **Leaderboard (MuSR)** | 0.05 | 0.22 | 0.15 | 10,979,309 | 1,731,217 |
| **MMLU** | 0.56 | 0.57 | 0.55 | 6,986,437 | 2,473,477 |
| **ARC** | 0.78 | 0.78 | 0.74 | 2,744,559 | 2,473,477 |
| **GSM8K** | 0.27 | 0.51 | 0.49 | 3,742,399 | 2,473,477 |
| **MTBench** | - | 5.00 | 4.97 | 9,232,227 | 2,473,477 |

The results, depicted in Table 12, indicate that the stacked training strategy achieves better performance than phased training across all benchmarks. Results in Table 13 indicate that the LAB hyperparameter configuration consistently outperforms TULU, reinforcing our earlier conclusion that larger batch sizes and a constant learning rate schedule are advantageous. These findings suggest that our recommended training strategies and hyperparameters are effective across different model architectures and sizes simultaneously, including the LLaMA family. Practitioners may consider applying these insights to fine-tune various small-sized LLMs, potentially achieving improvements in performance.

### A.6.9 EARLY TRAINING DYNAMICS AS PREDICTORS OF FINAL PERFORMANCE

In addition to the MTBench results presented in the main paper, we include the MMLU performance comparison here.

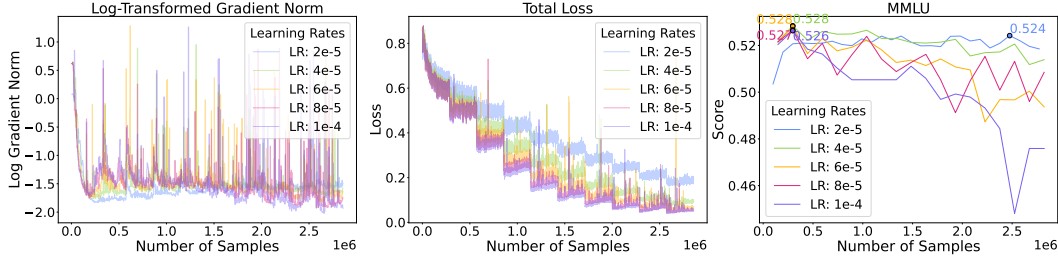

Figure 17: LAB Learning Rate Sweep: Impact on Training Dynamics (Grad Norm, Loss) and Final MMLU Performance.

Models trained with a learning rate of $2 \times 10^{-5}$ demonstrated lower gradient norms initially, which increased toward the end of training, and higher loss throughout, ultimately resulting in superior final performance on MMLU compared to models trained with higher learning rates. This pattern mirrors the observations made for MTBench in the main paper, reinforcing the correlation between early training dynamics and final performance across different benchmarks.

Figures 18, 13, and 19 illustrate the correlation between early training dynamics—gradient norms and loss values—and final benchmark performances across other experiments:

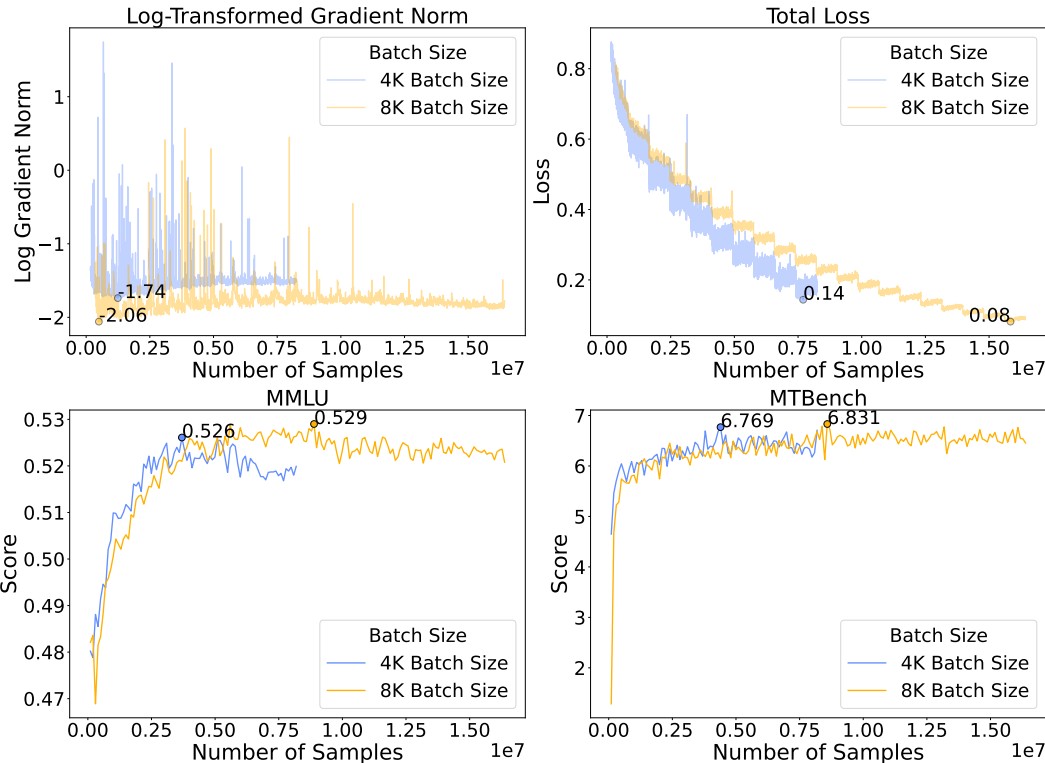

Figure 18: Effect of Batch Size (4k vs 8k) on Training Dynamics and Final Performance on MMLU and MTBench Benchmarks in Stacked Training.

- **Batch Size Comparison in Stacked Training.** We further examined the impact of batch size on early training dynamics and final performance in stacked training by comparing batch sizes of 3,840 and 7,680 samples (denoted as 4k and 8k). Figure 18 presents the gradient norms and loss values over training, along with the corresponding performances on MMLU and MTBench. We observed that the 8k batch size consistently exhibited lower gradient norms and higher loss throughout training compared to the 4k batch size. Ultimately, the 8k batch size achieved better final performance on both MMLU and MTBench benchmarks.

  The larger batch size likely benefits from increased data diversity within each batch, which reduces gradient variance and promotes stable updates. This diversity may help the model avoid large deviations from pre-trained parameters, allowing it to generalize more effectively while minimizing forgetting. However, we also noted that for smaller numbers of training samples, the 4k batch size achieved higher MMLU and MTBench scores, suggesting a trade-off. If computational resources or training time are limited, the 4k batch size may offer better performance in the early stages of training—up to approximately 3.75 million samples for MMLU and 8.5 million samples for MTBench. Beyond these points, the 8k batch size surpasses the 4k batch size in performance as observed in Figure 18.

- **Warmup Steps Comparison.** We examined the impact of different warmup configurations (0, 25, and 100 steps) on early training dynamics and final performance. All models demonstrated very similar performance, loss curves, and gradient norms throughout training. The model trained without warmup steps (0 warmup) achieved slightly better performance on the MMLU benchmark and comparable performance on MTBench compared to models trained with 25 or 100 warmup steps.

  Gradient norm and loss curves serve as a proxy for the smoothness of the optimization process. Large fluctuations in early gradnorm values may indicate instability, which could negatively affect convergence, while more stable or lower gradnorm magnitudes suggest a smoother path toward optimal performance. Given that all warmup configurations resulted in similar final performance across both benchmarks and exhibited nearly identical loss and gradnorm curves, it indicates a

strong correlation between training dynamics and final performance which can be seen in Figure 13.

- **Learning Rate Schedule Comparison.** We analyzed the effect of using a constant learning rate versus a cosine decay schedule with learning rates of $2 \times 10^{-5}$ and $4 \times 10^{-5}$ on early training dynamics and final performance. Figure 19 presents the gradient norms and loss values over training, along with the corresponding performances on MMLU and MTBench. Up until approximately 1 million samples, both learning rate schedules produce nearly identical gradient norms, loss values, and MMLU scores, as decay has not yet influenced the learning rate. After this point, while the cosine decay model shows slightly lower gradient norms and higher loss values, the constant learning rate configuration achieves comparable or better final performance on both MMLU and MTBench. The lower gradient norms with cosine decay suggest that the model is progressing in a stable direction within a flatter region of the loss landscape, indicating good generalization potential. Meanwhile, the higher loss values imply that the model is not overfitting to specific patterns in the data. However, the constant learning rate schedule maintains similar stability without compromising generalization, suggesting it may be more effective overall for these benchmarks.

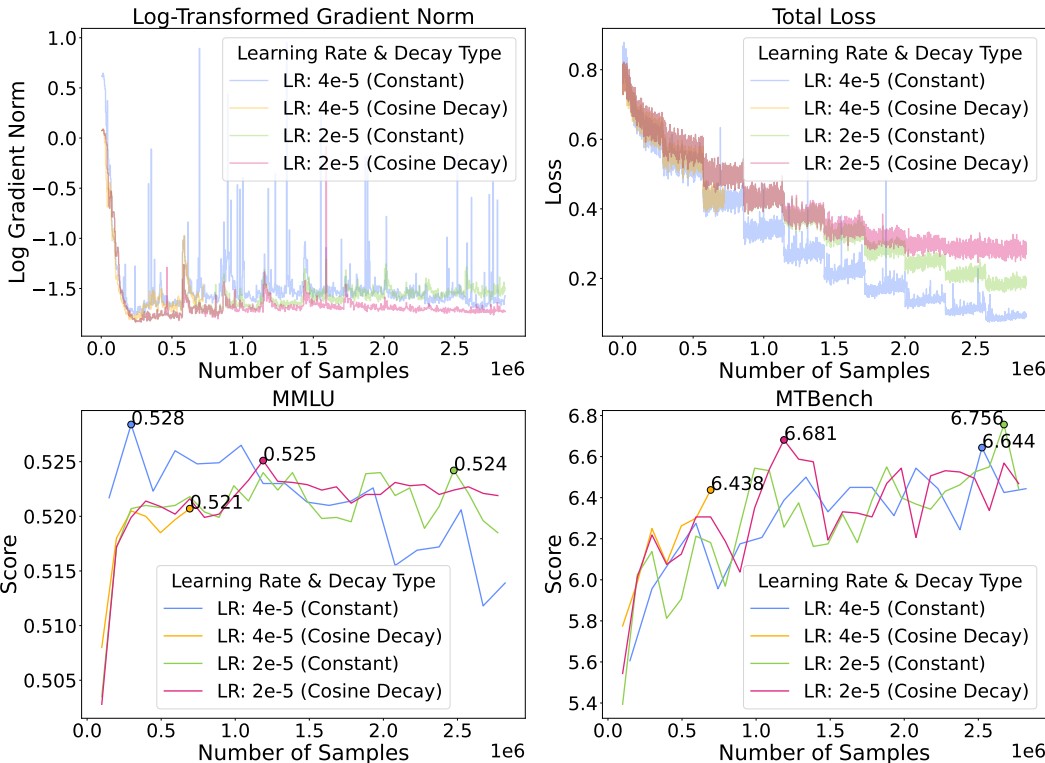

Figure 19: Effect of Constant Learning Rate vs. Cosine Decay Across Different Learning Rates on Training Dynamics (Grad Norm, Loss) and Final Performance on MMLU and MTBench Benchmarks.

### A.6.10 GRADIENT ACCUMULATION EQUIVALENCE TO FULL BATCH TRAINING

We investigated whether using gradient accumulation on a single node with a large batch size is equivalent to distributed training across multiple nodes with the same effective batch size. Theoretically, both methods should yield identical training dynamics and result in the same fine-tuned model if implemented correctly.

In our experiments, we compared two setups:

- **Single Node with Gradient Accumulation.** We utilized a single node with gradient accumulation to achieve an effective batch size corresponding to 60,000 tokens.

- **Multi-Node Distributed Training.** We employed distributed training across four nodes, maintaining the same effective batch size of 60,000 tokens without gradient accumulation.

We evaluated both setups by comparing their training loss curves, as well as performance on the MMLU and MTBench benchmarks. The results showed that the loss trajectories were virtually identical between the two methods. Additionally, the final performances on MMLU and MTBench were the same within experimental variance. These findings confirm that gradient accumulation on a single node can replicate the training dynamics and outcomes of full-batch distributed training across multiple nodes. This equivalence provides flexibility for practitioners with limited computational resources, allowing them to achieve the same model quality using gradient accumulation on fewer GPUs.

