# OpenReview forum: "Unveiling the Secret Recipe: A Guide For Supervised Fine-Tuning Small LLMs"
_ICLR.cc/2025/Conference — ICLR 2025 Poster_

### Official Review · Reviewer_V6Xk · 2024-10-31

**Soundness:** 3
**Presentation:** 3
**Contribution:** 2
**Rating:** 6
**Confidence:** 3

**Summary:**

The paper presents an in-depth study on fine-tuning small LLMs with 3B to 7B parameters using large-scale instruction tuning datasets across various knowledge domains and skills. The study challenges common training practices and offers new insights for customizing LLMs effectively. Key findings include the benefits of larger batch sizes with lower learning rates, the predictive value of early-stage training dynamics for final model performance, the lack of necessity for warmup phases, and the superiority of stacked training over phased training. These results provide an actionable and comprehensive guide for practitioners working on fine-tuning smaller LLMs.

**Strengths:**

S1: The paper gives practical guidance for fine-tuning small LLMs, which can be highly beneficial for researchers and practitioners.

S2: The experimental study is well-executed, covering a broad range of training configurations such as batch size, learning rate schedules, and training methods. It also tests three models against three datasets.

S3: The paper is well written. The related work section is comprehensive.

**Weaknesses:**

### Major weaknesses

W1: The paper doesn't examine the generalizability of the presented insights on new models and more domain-specific datasets. One of the motivations for using small LLMs is that practitioners can more effectively customize them for specific domains (L39). However, all the datasets used in the paper focus on general tasks (e.g., language understanding and STEM questions). To strengthen the paper's practical contributions, it would be beneficial to include training datasets and evaluation benchmarks from more specialized areas (e.g., legal documents, customer service, financial reports).

W2: Related to W1, the choice of the LLM models in the study seems arbitrary. Does the suggested training strategies work as well with the more popular small models (e.g., phi, llama)? The study could be improved by including these well-known open-source models.

W3: The main findings are not particularly surprising and somewhat expected (e.g., larger batch size improves performance, performance-efficiency trade-off).

### Minor weaknesses

M1: L37 needs citation for evidence.

**Questions:**

Q1: How well does the recommended training strategies work with the popular small LLMs (e.g., Phi, Llama)?

Q2: L38 argues that fine-tune smaller model is good for researchers with limited infrastructure support. However, L82 argues that to train a small model well, one needs a larger batch size and specified resources. Is this a contradiction?

---

> ### Author Response · Authors · 2024-11-21
> **Generalizability of findings to new models, evaluation on domain-specific datasets, discussion on empirical observations, and computational resource considerations.**
>
> We appreciate the reviewer’s careful reading of our paper and thoughtful comments!
>
> ---
> **Q1. Generalizability of the presented insights on new models.**
>
> A1. Thank you for highlighting this matter. In response, we have reproduced our experiments with the LLaMA 3B model, particularly focusing on phased v.s. stacked training and batch size impacts (TULU vs. LAB). Our findings align with those in our submission: larger batch sizes yield better benchmark performance (e.g., MMLU, MTBench, and Leaderboard), and stacked training is more sample-efficient and performs comparably to, or even better than, phased training.
>
> We also note that the Granite model shares the same architecture as the LLaMA model. Hence we believe that the findings in this paper can generalize across the LLaMA model family. In the paper, however, we focus on the Granite and Mistral model families due to their Apache-2.0 licensing, which is significantly more permissive than the LLaMA license. Our primary objective is to encourage collaborative LLM advancements within the open-source AI community by providing a detailed guide for practitioners working on fine-tuning smaller LLMs, which is why we selected the Granite models over LLaMA. We did not conduct experiments on the Phi model as it only releases an instruction-tuned version, whereas our study centers on instruction tuning for model customization starting from base models.
>
> ---
> **Q2. Domain-specific datasets.**
>
> A.2. Indeed, this is a great suggestion! To evaluate the generalizability of our findings to domain-specific datasets, we conducted experiments using a Math, Reasoning, and Code (MRC) dataset. This dataset specializes in mathematical problem-solving, logical reasoning, and programming tasks.
>
> We fine-tuned the LLaMA 3B model using both stacked and sequential phased training strategies with LAB hyperparameters, and we also compared the impact of batch size and the use of a constant learning rate (analyzed through TULU vs. LAB). For phased training, we partitioned the dataset into two phases based on response length, using length as a proxy for difficulty. We evaluated the models on several benchmarks, including GSM8K, ARC, MMLU, MTBench, and the Open LLM Leaderboard v2 benchmarks (MMLU-Pro, GPQA, MuSR, MATH, IFEval, and BBH). Our observation is consistent with our initial submission. For example, stacked training slightly outperforms phased training and is more sample-efficient across all benchmarks, and larger batch sizes (LAB) yield better benchmark performance (e.g., MMLU, MTBench, and Leaderboard).
>
> Finally, we remark that adapting a general-purpose LLM base model to be a domain specific expert involves multiple factors (e.g., domain-specific data, converting such data into instruction tuning datasets, base model selection, and integrating the data into the model). This paper focuses exclusively on the final step, offering a detailed guide for practitioners to determine training configurations.
>
> ---
> **Q3. Main findings are not surprising.**
>
> A.3. Yes, we agree—many insights, such as the performance improvement from larger batch sizes, are expected. That said, we note that some findings might challenge existing literature. For instance, hyperparameter recommendations from TULU are suboptimal in our experiments, and phased training as suggested by Orca does not benefit model performance compared to stacked training.
>
> Our objective is to share our empirical observations from fine-tuning LLMs. We support these insights through extensive experimentation, documenting effective hyperparameter settings and highlighting failure modes. We hope these results will offer valuable guidance to practitioners working with limited computational resources for hyperparameter tuning.
>
> ---
> **Q4. L37 needs citation for evidence.**
>
> A.4. Thanks for your suggestions! We added 3 citations to the revised paper.
>
> ---
> **Q5. L38 argues that fine-tune smaller model is good for researchers with limited infrastructure support. However, L82 argues that to train a small model well, one needs a larger batch size and specified resources. Is this a contradiction?**
>
> A.5. For researchers with limited computing resources, gradient accumulation offers a practical approach to achieve a large effective batch size. We confirm that gradient accumulation on a single node yields results comparable to those of multi-node distributed training; further details are provided in Appendix A.5.10.

---

> > ### Comment · Reviewer_V6Xk · 2024-11-26
> > **Thank you so much for the detailed response!**
> >
> > Thank you so much for the detailed response! It has addressed all my concerns. I have changed my score from 5 to 6. Thanks!

---

### Official Review · Reviewer_Y8PZ · 2024-11-03

**Soundness:** 3
**Presentation:** 4
**Contribution:** 3
**Rating:** 6
**Confidence:** 4

**Summary:**

This paper performs a deep investigation of the common practices used while fine-tuning small LLMs for domain specific downstream tasks. By conducting a range of experiments with Llama and Mistral models sized 3-7B, they provide a set of guidelines on the best practices to be used for instruction fine-tuning language models for specific tasks. Overall, using larger batch sizes with lower learning rates, a mixture of datasets rather than data curated phase-wise to prevent catastrophic forgetting and monitoring gradient norms and training loss during the initial stages of fine-tuning were shown to yield optimal performance across the tested models.

**Strengths:**

- The paper investigates an important and growing field of developing small LLMs for specialized domains to balance their task memeorization ability and generalization.
- The quality and design of experiments are good, and many commonly used techniques have been investigated, providing valuable guidance to practitioners.
- Overall, the paper is well-written and easy to understand. Though it has some limitations in terms of the breadth of experiments (mentioned below), and some results are already known, such as using a larger batch size and a lower learning rate -- I believe the community would still benefit from the findings of this work.

**Weaknesses:**

- It would have been nice to have a discussion on optimization techniques such as LoRA and the increasingly popular approximate optimization algorithms like GaLore as these are widely used while fine-tuning small LLMs.
- The authors could have included evaluations on harder datasets involving more reasoning, but this has been mentioned by them as a limitation.

**Questions:**

- The term "skill" is used quite loosely. What do you refer to as a skill - a specific domain like math or a reasoning like CoT?
- Phase 10 of complex skill development trains the model on tasks like poetry writing etc. - is this also organized as an instruction - answer task? If not, then I am curious to know how you handled different data styles (eg. instruction format QA and knowledge data from books etc.)

---

> ### Author Response · Authors · 2024-11-21
> **Clarifications on fine-tuning techniques, skill definitions, additional evaluations, and the structure of complex skill tasks as instruction-answer pairs.**
>
> We thank the reviewer for the thoughtful review and the kind comments!
>
> ---
> **Q1. Have a discussion on optimization techniques such as LoRA and the increasingly popular approximate optimization algorithms like GaLore.**
>
> A1. Indeed, this is a great suggestion. Our work primarily focused on supervised fine-tuning (SFT) as it aligns with our goal of customizing LLMs using knowledge and skills instruction tuning data. There is ongoing debate about whether parameter-efficient fine-tuning methods, such as LoRA, can help incorporate new knowledge into models [see e.g., Jiang etal., 2024]. Given this issue, we chose to focus exclusively on SFT, which is probably the most widely used way for instruction tuning and can maximize performance of small-sized LLMs. In the revised paper, we highlighted this limitation in Section 4 and clarified our setting (using SFT for instruction tuning) in Introduction.
>
> Reference.
>
> Jiang, Ting, et al. "MoRA: High-Rank Updating for Parameter-Efficient Fine-Tuning." arXiv preprint arXiv:2405.12130 (2024).
>
> ---
> **Q2. Evaluations on harder datasets.**
>
> A2. In response, we conducted additional evaluations on more benchmark datasets during the rebuttal period. Specifically, we evaluated our models on *MMLU-Pro, GPQA, MuSR, MATH, IFEval, and BBH* from the Open LLM Leaderboard v2, as well as *GSM8K and ARC*. These benchmarks cover a variety of tasks, including advanced reasoning, mathematical problem-solving, instruction following, and domain-specific knowledge. This broader evaluation helps to ensure that our findings are robust across different application areas and diverse contexts. For the specific experiment comparing the performance of the LAB setting against the TULU hyperparameter settings on the TULU dataset, we used the evaluation benchmarks from the TULU paper—*MMLU, GSM8K, BBH, ToxiGen, and TruthfulQA*. By using the same benchmarks as TULU, we can directly assess the impact of our hyperparameter choices relative to their results.
>
> Finally, we maintained the limitations of this work in Section 4 and cautioned readers that our results are intended solely as a reference for practitioners fine-tuning LLMs who are uncertain about how to initialize their experiments. Achieving optimal model performance requires tailoring the approach to specific use cases, evaluation metrics, base models, and other factors.
>
> ---
> **Q3. The term "skill" is used quite loosely. What do you refer to as a skill - a specific domain like math or a reasoning like CoT?**
>
> A3. In our paper, skills refer to the specific capabilities that the LLM needs for customization for downstream use cases. Specifically, the dataset we used includes foundational skills (data from mathematics, coding, and reasoning, linguistic ability) and compositional skills (tasks that require a combination of knowledge and foundational skills).
>
> In response to your question, we added the following paragraphs to the revised paper to elaborate on the curation process for the knowledge and skills data.
>
> The datasets were curated using a taxonomy-driven approach to ensure comprehensive coverage of instruction-following, foundational knowledge, and compositional skills. The taxonomy hierarchically organizes tasks into three main branches—knowledge, foundational skills, and compositional skills—each further divided into granular subcategories. For each subcategory, manually written instruction-response pairs served as seed examples. These examples guided synthetic data generation using teacher models (e.g., Mixtral-7x8B) to expand the dataset while maintaining high quality and diversity. For knowledge data, reliable sources such as textbooks and technical manuals provided a grounding for synthetic questions and responses. Foundational skills data were drawn from public datasets covering essential areas like mathematics, coding, and reasoning. Compositional skills were synthesized using a taxonomy-guided approach to combine knowledge and foundational skills for complex tasks, such as writing detailed emails or generating logical arguments.
>
> ---
> **Q4. Phase 10 of complex skill development trains the model on tasks like poetry writing etc. - is this also organized as an instruction - answer task?**
>
> A4. Yes, it is organized as an instruction-answer task. An example is:
>
> Instruction: "Write a haiku that captures the beauty of a single autumn leaf."
> Answer: "Crimson leaf adrift,\nDancing on a cool fall breeze,\nNature's art takes flight."

---

### Official Review · Reviewer_VvP5 · 2024-11-03

**Soundness:** 3
**Presentation:** 3
**Contribution:** 2
**Rating:** 6
**Confidence:** 3

**Summary:**

This paper studies how to fine-tune small-size LLMs on large-scale instruction-tuning datasets effectively, measuring downstream performance on MMLU and MTBench. The experiments use Granite 3B, 7B, and Mistral 7B as the base models and compare stacked versus phased training, varying batch sizes, learning rates, and training configurations. The authors find that stacked training is preferable, a trade-off between batch size, and the possibility of omitting warm-up steps without sacrificing model quality.

**Strengths:**

- The investigation is well-motivated given the increasing interest in customizing general LLMs for domain-specific tasks. Understanding the design decisions and potential trade-offs is an important area of study.
- The writing was also generally easy to follow.

**Weaknesses:**

The experimental setup could be better justified. It was unclear what order of experiments the authors ran and to what extent are the findings dependent on this ordering. It seems like the authors did not do a full sweep of the entire cross-product of experimental conditions (if so, please present the full set of results for generality). Furthermore, other natural questions are:
- Why these choices of models?
- How were the fine-tuning datasets curated? This was not discussed in Section 2.1 or A.2. How might results vary depending on whether the data you are trying to fine-tune is more general or specific? This is related to the motivating telecommunications example.
- Why are MMLU and MTBench the right datasets for evaluation?
- Why were TULU vs LAB used as primary comparisons for hyperparameter configurations?

The presentation of experimental results could be significantly improved.
- Showing individual training curves is quite noisy, could the authors run with multiple seeds and present smoother results?
- Further, why did the authors choose not to present most results in a table? This way it would be much easier to capture findings across all three models.

Clarity on claims and significance of results.
- There is generally a lack of baselines in this work. It’s important to note what are baseline performance of the various Granite and Mistral models on MMLU an MTBench.
- Relatedly, the strength of the claimed results could be clarified. For example, many of the reported numbers are very close: 6.77 vs 6.76 on MTBench in Figure 1a, 0.5251 vs 0.5242 in Figure 3a, and 0.59 vs 0.6 vs 0.61 in Figure 4a? Having some baseline values would help readers determine whether these are significant differences.

**Questions:**

Please address the specific questions raised in the weaknesses section.

The revised presentation of the work addresses many of my concerns and I have increased my score accordingly.

---

> ### Author Response · Authors · 2024-11-21
> **Clarifications on experimental order, model and dataset choices, evaluation benchmarks, and added baseline results in response to reviewer feedback.**
>
> We thank the reviewer for the thoughtful comments and for appreciating the novelty of the work!
>
> ---
> **Q1. What order of experiments the authors ran and to what extent are the findings dependent on this ordering?**
>
> A1. Indeed, this is a great suggestion! In response, we clarified the specific experimental configurations in Section 2.3 (Line 216–236). In short, we used the LAB hyperparameter configuration as the default for all experiments where we varied a single factor
> (e.g., batch size, learning rate, learning rate schedule, training strategy) while keeping all other
> settings constant to isolate its effect. We started with a cross-product of learning rates and batch sizes (e.g., low and high values for each) to identify stable configurations. Using the best settings, we explored phased vs. stacked training by conducting a batch size sweep (128, 4K, 8K) and then assessed the effect of using a constant learning rate.
>
> ---
> **Q2. Why these choices of models?**
>
> A2. Our experiments focus on the Granite and Mistral model families. We select these models due to their Apache-2.0 license, which is one of the most permissive licenses available for off-the-shelf LLMs. This choice aligns with our goal to foster collaborative LLM development in the open-source AI community by providing a detailed guide for practitioners focused on fine-tuning smaller models. We note that the Granite model shares the same architecture as the LLaMA model. Hence we believe that the findings in this paper can generalize across the LLaMA model family. In the paper, however, we focus on the Granite and Mistral model families since Apache-2.0 licensing is significantly more permissive than the LLaMA license. During rebuttal, we have reproduced our experiments with the LLaMA 3B model, particularly focusing on phased v.s. stacked training and batch size impacts (TULU vs. LAB). Our findings align with those in our submission: larger batch sizes yield better benchmark performance (e.g., MMLU, MTBench, and Leaderboard), and stacked training is more sample-efficient and performs comparably to, or even better than, phased training.
>
> ---
> **Q3. How were the fine-tuning datasets curated? How might results vary depending on whether the data is more general or specific?**
>
> A3. The datasets were curated using a taxonomy-driven approach to ensure comprehensive coverage of instruction-following, foundational knowledge, and compositional skills. The taxonomy hierarchically organizes tasks into three main branches—knowledge, foundational skills, and compositional skills—each further divided into granular subcategories. For each subcategory, manually written instruction-response pairs served as seed examples. These examples guided synthetic data generation using teacher models (e.g., Mixtral-7x8B) to expand the dataset while maintaining high quality and diversity. For knowledge data, reliable sources such as textbooks and technical manuals provided a grounding for synthetic questions and responses. Foundational skills data were drawn from public datasets covering essential areas like mathematics, coding, and reasoning. Compositional skills were synthesized using a taxonomy-guided approach to combine knowledge and foundational skills for complex tasks, such as writing detailed emails or generating logical arguments. This structured curation ensures that this dataset is representative of real-world use cases for model customization.
>
> This dataset represents a balance of general and domain-specific data, reflecting scenarios where models must generalize broadly while also adapting to specific tasks. This diversity ensures our findings—on the impact of batch size, learning rates, and training strategies—are applicable to other datasets with similar generality and specialization balance. For highly specific datasets, the observed trends might vary slightly, but the training dynamics we analyze, such as gradient norms and loss behaviors, provide a robust framework to identify optimal settings even in those cases. During the rebuttal period, to validate the generalizability of our findings, we included results on a dataset covering math, reasoning, and coding tasks, focusing on comparisons between phased and stacked training, as well as the impact of batch size and the use of a constant learning rate (analyzed through TULU vs. LAB).

---

> ### Author Response · Authors · 2024-11-21
> **Clarifications on experimental order, model and dataset choices, evaluation benchmarks, and added baseline results in response to reviewer feedback.**
>
> We thank the reviewer for the thoughtful comments and for appreciating the novelty of the work!
>
> ---
> **Q4. Why are MMLU and MTBench the right datasets for evaluation?**
>
> A4. We selected MMLU and MTBench because they provide diverse coverage for general knowledge, conversational abilities, task-specific dialogue adaptations, and reasoning across a wide range of topics. These features are essential for model customization and closely align with the central question of our study: how to fine-tune a small-size LLM on large-scale instruction tuning datasets that cover diverse knowledge and skills.
>
> During the rebuttal periods, we conducted additional evaluations on more benchmark datasets. Specifically, we evaluated our models on *MMLU-Pro, GPQA, MuSR, MATH, IFEval, and BBH* from the Open LLM Leaderboard v2, as well as *GSM8K and ARC*. These benchmarks cover a variety of tasks, including advanced reasoning, mathematical problem-solving, instruction following, and domain-specific knowledge. This broader evaluation helps to ensure that our findings are robust across different application areas and diverse contexts. For the specific experiment comparing the performance of the LAB setting against the TULU hyperparameter settings on the TULU dataset, we used the evaluation benchmarks from the TULU paper—*MMLU, GSM8K, BBH, ToxiGen, and TruthfulQA*. By using the same benchmarks as TULU, we can directly assess the impact of our hyperparameter choices relative to their results.
>
> ---
> **Q5. Why were TULU vs LAB used as primary comparisons for hyperparameter configurations?**
>
> A5. We selected TULU and LAB as primary comparisons for hyperparameter configurations due to their prominence and alignment with our study's focus. TULU is widely regarded as a gold standard for fine-tuning LLMs, leveraging high-quality instruction datasets and a two-stage training approach (instruction tuning followed by preference tuning). LAB, on the other hand, uses similar knowledge and skills data but introduces a multi-phase tuning framework aimed at reducing reliance on human annotations. LAB also advocates for phased training, which we found to be suboptimal compared to stacked training. By comparing these configurations, we systematically evaluated widely accepted practices, challenged their limitations, and provided actionable guidelines for improving performance and efficiency in fine-tuning small LLMs.
>
> ---
> **Q6. Showing individual training curves is quite noisy.**
>
> A6. We presented individual training curves due to computation constraints. We acknowledge the potential noise in individual training curves and added a discussion about this issue in the limitations section of the revised paper. Similar challenges have been noted in prior work on LLM fine-tuning, such as TULU and LAB, which also study hyperparameter tuning but do not consistently use multiple seeds due to computational constraints. Each phase of our training requires approximately 10 hours on 80 A100 GPUs, making repeated runs for multiple seeds prohibitively expensive. However, the stability observed in training dynamics (e.g., loss curves, gradient norms) across time suggests that the results are representative and unlikely to diverge significantly. This informed our decision to proceed with single-seed runs for the reported experiments.
>
> ---
> **Q7. Present most results in a table.**
>
> A7. Thanks for your suggestion! In response, we have replaced most figures with tables in the revised paper, relocating the original figures to the appendix for reference.
>
> ---
> **Q8. What are baseline performance of the various Granite and Mistral models on MMLU and MTBench?**
>
> A8. Thank you for the insightful suggestion. We agree that adding baseline scores for the pretrained base Granite and Mistral models on MMLU and MTBench improves clarity. These values are now included in the revised paper to better contextualize the fine-tuning improvements and highlight the significance of the reported metrics.

---

> > ### Comment · Reviewer_VvP5 · 2024-11-25
> >
> > Thank you for carefully responding to the points I raised. I really appreciate the table format of the results compared to the old plots and have raised my score accordingly.

---

### Official Review · Reviewer_WF61 · 2024-11-04

**Soundness:** 3
**Presentation:** 3
**Contribution:** 2
**Rating:** 6
**Confidence:** 3

**Summary:**

This paper explores methods for making smaller language models (3B–7B parameters) effective in specific domains without requiring extensive computing power. It focuses on optimizing model fine-tuning through adjustments in batch size, learning rates, and training approaches. The authors compare two main training strategies: stacked training, where models learn from varied data all at once, and phased training, where data is introduced in stages. Their findings show that stacked training is both faster and more efficient. Additionally, using larger batch sizes and lower learning rates enhances model performance on benchmarks. The research offers practical tips, such as skipping warmup phases and using a constant learning rate, which simplify the training process. This guide provides useful insights for developers and organizations that want to adapt small language models effectively for specialized tasks on limited resources, making advanced AI more accessible.

**Strengths:**

1.For the fine-tuning stage of small-scale LLM instructions, by utilizing different types of data and exploring various parameter settings and data organization methods, comprehensive experimental conclusions have been obtained, especially for the setting of large batch size, which is often easily achievable in real-world scenarios through gradient accumulation techniques.
2.By evaluating stacked versus phased training approaches, the paper demonstrates that stacked training is generally more efficient, saving time and resources. This finding is especially beneficial for practitioners looking to balance performance with computing constraints.
3.The final conclusion drawn in the paper challenges widely recognized practices such as TULU.

**Weaknesses:**

1.The study is focused on small models (3B–7B parameters) and may not generalize to larger models. As such, its findings might not apply to larger language models often used in advanced applications, which limits the scope of the conclusions.
2.The experiments mainly use Granite and Mistral model families, leaving out other architectures. This limits the ability to generalize the results to models with different foundational architectures or pre-training techniques.  I suggest adding the llama3.2 series (1B, 3B) as well as the llama3.1-8B model when resources permit.
3.The evaluation relies primarily on MMLU and MTBench benchmarks, which, while broad, may not represent all potential application areas. This leaves open the question of whether the findings would hold on other benchmarks like GSM8K or ARC, which focus on different types of reasoning and knowledge.
4. There are some details in the paper that need further checking, such as the repetition of the citation to the paper Lab: Large-scale alignment for chatbots.

**Questions:**

See weeknesses

---

> ### Author Response · Authors · 2024-11-21
> **Clarifications and additional experiments addressing the generalizability of findings, evaluation benchmarks, and architectural coverage for small LLMs.**
>
> We thank the reviewer for the thoughtful review and for appreciating the merits of the work!
>
> ---
> **Q1. The study is focused on small models and may not generalize to larger models.**
>
> A1. Our focus is on small-sized LLMs (3B–7B parameters) as they are frequently used by individual researchers and smaller organizations as the backbone for model customization. We acknowledge that some findings may not extend to larger models and have discussed this limitation in Section 4. Nonetheless, we believe this paper can be a valuable resource for developers with limited resources, as small-sized LLMs are significantly more accessible in terms of both fine-tuning and inference costs. Additionally, we hope it fosters a collaborative environment within the open-source community, empowering more contributors to participate in LLM research and training, extending LLM development beyond large organizations.
>
> ---
> **Q2. The experiments mainly use Granite and Mistral model families, leaving out other architectures.**
>
> A2. Thank you for highlighting this matter. In response, we have reproduced our experiments with the LLaMA 3B model, particularly focusing on phased v.s. stacked training and batch size impacts (TULU vs. LAB). Our findings align with those in our submission: larger batch sizes yield better benchmark performance (e.g., MMLU, MTBench, and Leaderboard), and stacked training is more sample-efficient and performs comparably to, or even better than, phased training.
>
> We also note that the Granite model shares the same architecture as the LLaMA model. Hence we believe that the findings in this paper can generalize across the LLaMA model family. In the paper, however, we focus on the Granite and Mistral model families due to their Apache-2.0 licensing, which is significantly more permissive than the LLaMA license. Our primary objective is to encourage collaborative LLM advancements within the open-source AI community by providing a detailed guide for practitioners working on fine-tuning smaller LLMs, which is why we selected the Granite models over LLaMA.
>
> ---
> **Q3. The evaluation relies primarily on MMLU and MTBench benchmarks, which, while broad, may not represent all potential application areas.**
>
> A3. We selected MMLU and MTBench because they provide diverse coverage for general knowledge, conversational abilities, task-specific dialogue adaptations, and reasoning across a wide range of topics. These benchmarks align well with the central question of our study: how to fine-tune a small-size LLM on large-scale instruction tuning datasets that cover diverse knowledge and skills.
>
> In response to your concerns, we conducted additional evaluations on more benchmark datasets during the rebuttal period. Specifically, we evaluated our models on *MMLU-Pro, GPQA, MuSR, MATH, IFEval, and BBH* from the Open LLM Leaderboard v2, as well as *GSM8K and ARC* as you recommended. These benchmarks cover a variety of tasks, including advanced reasoning, mathematical problem-solving, instruction following, and domain-specific knowledge. This broader evaluation helps to ensure that our findings are robust across different application areas and diverse contexts. For the specific experiment comparing the performance of the LAB setting against the TULU hyperparameter settings on the TULU dataset, we used the evaluation benchmarks from the TULU paper—*MMLU, GSM8K, BBH, ToxiGen, and TruthfulQA*. By using the same benchmarks as TULU, we can directly assess the impact of our hyperparameter choices relative to their results. Finally, we have included a discussion on these limitations in Section 4.
>
> ---
> **Q4.  There are some details in the paper that need further checking.**
>
> A4. Thanks for your careful reading of our paper! We corrected these citations in the revised paper.

---

### Author Response · Authors · 2024-11-25
**Summary of New Experiments and Updates During Rebuttal**

We thank all Reviewers for their time and effort! We are glad to see our paper was positively received and strove to address the key points and questions raised by each Reviewer. We also recognize that the Reviewers are busy handling multiple papers, so their thoughtful feedback is even more appreciated. Below, we outline the main changes and additional experiments incorporated during the discussion period:

- ### **Generalization to the LLaMA Model Family (Refer to Lines 1399–1476):**
    We conducted new experiments using the LLaMA 3B model, focusing on comparisons between LAB and TULU hyperparameters and stacked vs. phased training strategies. The findings align with our original submission, reaffirming that larger batch sizes and stacked training provide better performance and stacked training is more sample efficient. In our response to reviewers, we clarified why Granite and Mistral models were the primary focus in the paper, due to their permissive licensing.

- ### **Evaluation on Domain-Specific Data (Refer to Lines 1223–1295):**
    To assess generalizability to domain-specific datasets, we fine-tuned models on a Math, Reasoning, and Code (MRC) dataset. LAB hyperparameters and stacked training consistently outperformed TULU and phased training across benchmarks like GSM8K, ARC, and others.

- ### **Cross-Dataset Evaluation with TULU (Refer to Lines 378–402) and New Difficulty-Based Phased Training (Refer to Lines 969–1015):**
    We conducted experiments on the TULU dataset to test the robustness of larger batch sizes and constant learning rate, confirming that LAB (4k batch size) outperforms TULU (128 batch size). Additionally, we tested a new phased training approach inspired by recent prior works [see e.g., Pang et al., 2024; Mitra et al., 2023], where the dataset is partitioned based on difficulty using response length as a proxy. Our findings indicate no performance gains compared to stacked training.

- ### **Expanded Benchmarks:**
    In addition to MMLU and MTBench, evaluations were conducted on GSM8K, ARC, Open LLM Leaderboard v2 benchmarks, such as MMLU-Pro, GPQA, MuSR, MATH, IFEval, and BBH, as well as TULU-specific benchmarks like TruthfulQA and ToxiGen. The results confirm the robustness of our findings across diverse evaluation settings.

- ### **Improved Presentation:**
    Figures were replaced with tables for clarity, and corrections were made to citations and other minor typos. Results are now more accessible and better contextualized with baseline performance metrics.

Our observations from these additional experiments are consistent with the four main takeaways presented in our initial submission. Please do follow up with us if you have additional suggestions and feedback that can further strengthen the paper. Thank you!

---

### **References:**

- Pang, Wei, et al. *"Phased Instruction Fine-Tuning for Large Language Models."* arXiv preprint arXiv:2406.04371 (2024). [https://arxiv.org/abs/2406.04371](https://arxiv.org/abs/2406.04371)

- Mitra, Arindam, et al. *"Orca 2: Teaching Small Language Models How to Reason."* arXiv preprint arXiv:2311.11045 (2023). [https://arxiv.org/abs/2311.11045](https://arxiv.org/abs/2311.11045)

---

### Meta-Review · Area_Chair_LESH · 2024-12-20

**Metareview:**

This paper addresses the customization of LLMs by focusing on effective training strategies for fine-tuning models with 3B to 7B parameters using large-scale instruction tuning datasets across diverse knowledge domains. Through an in-depth study of various training configurations on three pretrained LLMs, the results challenge several common training practices, including hyperparameter recommendations from TULU and phased training suggested by Orca. The paper is well-written, and the authors addressed almost all of the reviewers' concerns.

**Additional Comments On Reviewer Discussion:**

The paper is well-written, and the authors addressed almost all of the reviewers' concerns. Reviewers VvP5 and V6Xk raised their rating from 5 to 6.

---

### Decision · Program_Chairs · 2025-01-22

Accept (Poster)